# SMARCA4 deficient tumours are vulnerable to KDM6A/UTX and KDM6B/JMJD3 blockade

Octavio A. Romero [1,11✉], Andrea Vilarrubi [1,11], Juan J. Alburquerque-Bejar [1], Antonio Gomez [2], Alvaro Andrades[3,4], Deborah Trastulli [5], Eva Pros [1], Fernando Setien[1], Sara Verdura[5], Lourdes Farré [6], Juan F. Martín-Tejera[6], Paula Llabata [1], Ana Oaknin [7], Maria Saigi [1,8], Josep M. Piulats[8], Xavier Matias-Guiu[9], Pedro P. Medina [3,4], August Vidal [9,10], Alberto Villanueva [6,10] & Montse Sanchez-Cespedes [1✉]

Despite the genetic inactivation of *SMARCA4*, a core component of the SWI/SNF-complex commonly found in cancer, there are no therapies that effectively target SMARCA4-deficient tumours. Here, we show that, unlike the cells with activated *MYC* oncogene, cells with *SMARCA4* inactivation are refractory to the histone deacetylase inhibitor, SAHA, leading to the aberrant accumulation of H3K27me3. *SMARCA4*-mutant cells also show an impaired transactivation and significantly reduced levels of the histone demethylases KDM6A/UTX and KDM6B/JMJD3, and a strong dependency on these histone demethylases, so that its inhibition compromises cell viability. Administering the KDM6 inhibitor GSK-J4 to mice orthotopically implanted with *SMARCA4*-mutant lung cancer cells or primary small cell carcinoma of the ovary, hypercalcaemic type (SCCOHT), had strong anti-tumour effects. In this work we highlight the vulnerability of KDM6 inhibitors as a characteristic that could be exploited for treating *SMARCA4*-mutant cancer patients.

[1] Cancer Genetics Group, Josep Carreras Leukaemia Research Institute (IJC), Badalona, Barcelona, Spain. [2] Rheumatology Research Group, Vall d'Hebron Research Institute, Barcelona, Spain. [3] Department of Biochemistry and Molecular Biology I. Faculty of Sciences, University of Granada, Granada, Spain. [4] GENYO. Centre for Genomics and Oncological Research: Pfizer, University of Granada, Andalusian Regional Government, Granada, Spain. [5] Genes and Cancer Group, Cancer Epigenetics and Biology Program (PEBC), Bellvitge Biomedical Research Institute-IDIBELL Barcelona, Barcelona, Spain. [6] Chemoresistance and Predictive Factors Group, Program Against Cancer Therapeutic Resistance (ProCURE), Catalan Institute of Oncology (ICO), Oncobell Program, Bellvitge Biomedical Research Institute (IDIBELL), L'Hospitalet del Llobregat, Barcelona, Spain. [7] Department of Medical Oncology, Vall d'Hebrón Hospital, Barcelona, Spain. [8] Department of Medical Oncology, Catalan Institute of Oncology (ICO), Barcelona, Spain. [9] Department of Pathology, University Hospital of Bellvitge, IDIBELL, CIBERONC, L'Hospitalet del Llobregat, Barcelona, Spain. [10] Xenopat S.L., Parc Científic de Barcelona (PCB), Barcelona, Spain. [11] These authors contributed equally: Octavio A. Romero, Andrea Vilarrubi. ✉email: oromero@carrerasresearch.org; mscespedes@carrerasresearch.org

Chromatin remodelling is one of the epigenetic processes that is commonly disturbed in cancer, mainly through alterations in the mammalian switch/sucrose non-fermentable (SWI/SNF) complex. This complex modifies the structure of the chromatin by the ATP-dependent disruption of DNA–histone interactions at the nucleosomes, thereby activating or repressing gene expression. The various functions and components of the SWI/SNF complex have been thoroughly reviewed elsewhere[1,2].

Alterations at genes encoding different components of the SWI/SNF complex are present in a variety of tumour types and are thus an important feature of cancer development[3]. The *SMARCA4* (also known as *BRG1*) gene codes a core catalytic component of the SWI/SNF complex that features a bromodomain and helicase/ATPase activity[1,2]. Our own previous work produced the first evidence that *SMARCA4* is genetically inactivated in cancer and that SMARCA4 deficiency prevents the response to pro-differentiation stimuli in cancer cells[4–6]. In lung cancer, *SMARCA4* inactivation affects about one-third of non-small cell lung cancers (NSCLCs) and preferentially occurs against a background of wild type MYC (either C, L or N) or of members of the MYC-axis, such as MAX or MGA[4–7]. This hints at the existence of an important network that connects SWI/SNF and MAX/MYC functions. Mutations of *SMARCA4* also occur in other types of cancer, notably in the rare and very aggressive small cell carcinoma of the ovary, hypercalcaemic type (SCCOHT)[8], in which *SMARCA4* inactivation has been reported in almost 100% of cases[9–11].

The progress made towards understanding the role of chromatin remodelling in cancer development highlights the great potential of new epigenetic-based therapeutic strategies. With particular reference to SMARCA4, some previous studies have sought the vulnerabilities of SMARCA4-deficient tumours with a view to exploiting them for cancer treatment. SMARCA4 and SMARCA2 are mutually exclusive catalytic subunits of the SWI/SNF complex, and the inhibition of SMARCA2 activity appears to be synthetic lethal in cancer cells carrying *SMARCA4*-inactivating mutations, an effect that could be explained by paralogue insufficiency[12,13]. Further, SWI/SNF-mutant cancer cells with a wild type *KRAS* background depend on the non-catalytic action of the histone methyltransferase, EZH2[14]. However, we currently know of no small compounds that are capable of suppressing the ATPase or non-catalytic functions of SMARCA2 and EZH2, respectively, so these molecules are not yet suitable for use in therapeutic interventions. More recently, it has been proposed that cancer cells with an inactive SMARCA4 may be susceptible to CDK4/6 inhibitors[15].

On the other hand, components of the SWI/SNF complex bind to various nuclear receptors (e.g., oestrogen, androgen, glucocorticoid and retinoid receptors), thereby adapting the gene expression programmes to the demands of the cell environment[16–19]. We have reported that SMARCA4 is required to promote cell growth inhibition triggered by corticoids and retinoids in cancer cells[6] and that such effects are enhanced by combination with the pan- histone deacetylase (HDAC) inhibitor suberanilohydroxamic acid (SAHA)[20]. We observed that *MYC*-amplified but not *SMARCA4*-mutant cancer cells were sensitive to these treatments.

Here, we show that cancer cells that lack SMARCA4 have a defective regulation of H3K27ac/H3K27me3 and exhibit low levels of KDM6s, indicative of a deficient activity of these demethylases. This forms not only the basis of the refractoriness to SAHA, but also sensitises the cancer cells to inhibition of the demethylase activity of KDM6s, heavily compromising their viability.

## Results

### Refractoriness to growth inhibition and increase H3K27me3 by SAHA in SMARCA4def cells.
We studied the differential response to SAHA in lung cancer cells with oncogenic activation of MYC (hereafter referred to as MYCamp) with respect to those with genetic inactivation of SMARCA4 (hereafter, SMARCA4-def). The administration of SAHA was more effective at reducing the growth of MYCamp cells than of SMARCA4def cells (mean of half-maximum effective concentrations ($EC_{50}$) of 0.5 and 1.4 μM, for each group, respectively) (Fig. 1a, b, Supplementary Fig. 1a). Flow cytometric analysis showed that, in the MYCamp cells, SAHA blocked cell cycle progression at the G0/G1 or SubG1 phases and increased apoptosis, whereas no changes were observed in SMARCA4def cells (Supplementary Fig. 1b–d; Supplementary Fig. 2). These effects were not influenced by lung cancer histopathological subtypes, since they occurred in NSCLC and SCLC types. The selective sensitivity to SAHA of MYCamp cells was validated using publicly available datasets including more than 750 cancer cell lines of different origin and genetic background (Fig. 1c).

SAHA increases global histone acetylation, favouring an open chromatin structure and promoting transcriptional activation[21]. Transcriptionally active chromatin domains are characterised by a distinct array of histone marks, e.g., H3K27ac, H3K4me1 and H3K4me3, whereas H3K27me3 and H2AK119ub are often found at silent gene loci[22,23]. We first tested the effects of SAHA in global H3K27ac and H3K27me3, two different marks on the same residue, but with opposite functions[22,23]. As expected, SAHA triggered an increase in global H3K27ac in all the cells, while we observed an aberrant accumulation of H3K27me3 in the SMARCA4def cells, rather than the expected decrease (Fig. 1d). To determine whether this was due to a defective SMARCA4, we used H1299 cells, which lack SMARCA4 expression owing to an intragenic homozygous deletion[4], and restored wild type SMARCA4 (H1299-wtSMARCA4), using a doxycycline-inducible system, as previously reported[6]. As a control, we expressed a mutant form that lacked the ATPase domain (p.Glu668_Gln758del)[4] (hereafter referred to as H1299-mutSMARCA4). Administration of SAHA did not affect global H3K27me3 in the H1299-wtSMARCA4 cells, whereas the H1299-mutSMARCA4 cells underwent an increase in H3K27me3 concomitantly with a decrease in global H3K27ac (Fig. 1e). A dominant negative function of an overexpressed mutant SMARCA4 protein may underlie this effect in the H1299-mutSMARCA4 cells. Thus, the absence of a functional SMARCA4 induces defects in the dynamics of the H3K27me3 mark, following administration of SAHA.

The net levels of H3K27me3 are dictated by the coordinated action of histone methyltransferases (EZH2) and demethylases (KDM6A and KDM6B)[23]. The administration of SAHA did not alter the levels of EZH2 in most SMARCA4def cells (except in the H23 cells which showed an increase), indicating that overactivation of the methyltransferase activity of EZH2 is unlikely to account for the defects in H3K27me3 triggered by SAHA in these cells (Fig. 1f; Supplementary Fig. 3a). In contrast, treatment with SAHA alone reduced EZH2 levels in the MYCamp cells, consistent with the fact that SAHA triggers inhibition of cell cycle progression and increases apoptosis in cancer cells with this genetic context. These effects were enhanced by the addition of GSK126, an inhibitor of the enzymatic activity of EZH2. Previous studies have shown that GSK126, alone or combined with HDAC inhibitors, suppress the growth of the small cell carcinoma of the ovary hypercalcaemic type (SCCOHT)[24,25]. Here, the administration of GSK126, alone or in combination with SAHA, did not reduce the proliferation or viability of the SMARCA4def lung cancer cells (Supplementary Fig. 3b–d), implying that EZH2 activity does not, by itself, cause the refractoriness to cell growth inhibition by SAHA in SMARCA4def lung cancer cells.

### Regulation of KDM6 expression by SMARCA4.
The histone demethylases KDM6A (also known as UTX) and KDM6B (also

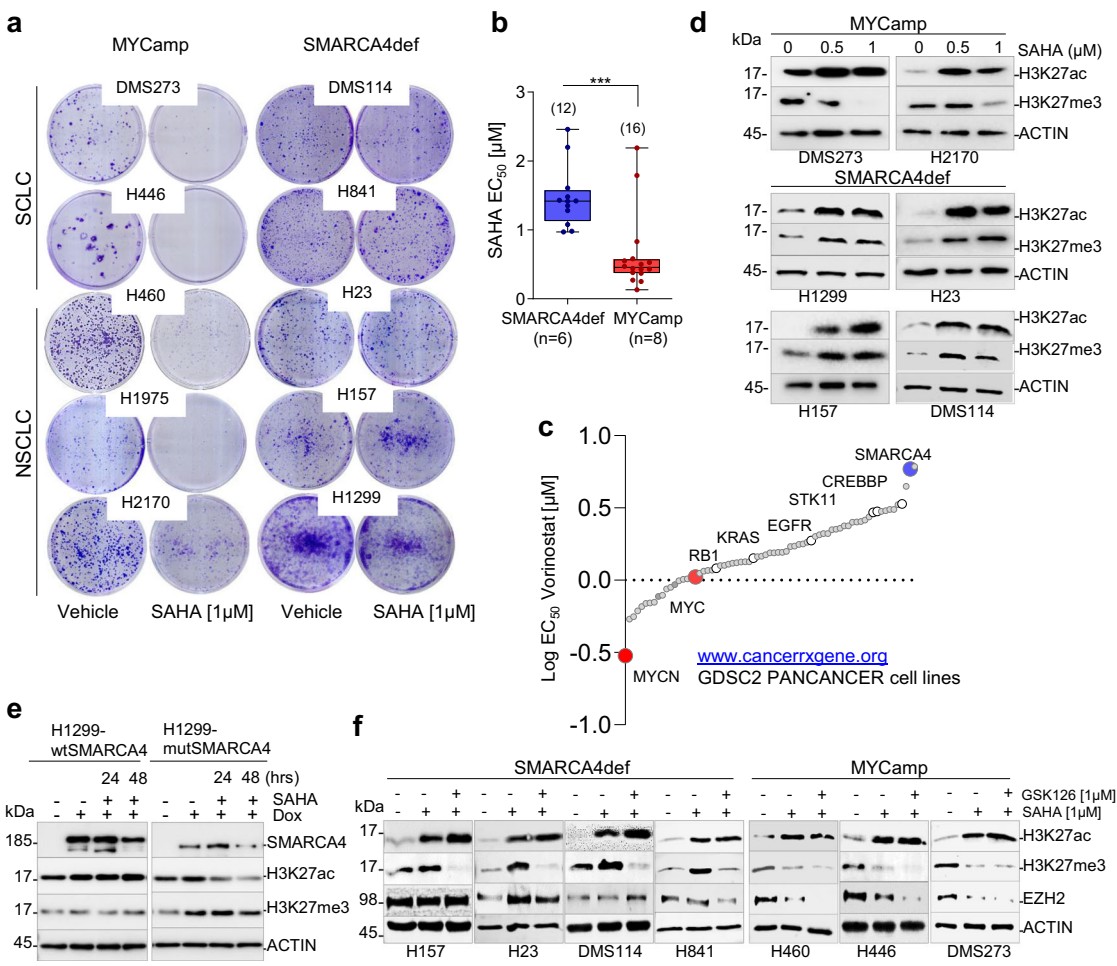

**Fig. 1 SAHA reduces growth and H3K27me3 levels in MYCamp, but not in SMARCA4def cancer cells. a** Clonogenic assays of the indicated lung cancer cells, untreated (vehicle) or treated with SAHA. NSCLC, non-small cell lung cancer; SCLC small cell lung cancer. **b** Distribution and mean of $EC_{50}$ values for treatment with SAHA in MYCamp (0.5 μM) and SMARCA4def cells (1.4 μM) (from MTT assays for cell viability; Supplementary Fig. 1). Bars show mean ± SD; two-sided unpaired Student's *t*-test. Values from two independent experiments per cell line are represented. In box-whisker plots, the horizontal band inside box indicates the median, the bottom and top edges of the box 25th–75th percentiles and the whiskers indicate the min to max. \*\*\*$P = 0.0002$. **c** Plot with the comparative half maximal effective concentration ($EC_{50}$) values for treatment with SAHA (vorinostat) in a panel of 758 cancer cell lines (from www.cancerrxgene.org database), according to the presence of selected gene alterations, including amplification of *MYC* oncogenes and inactivation of *SMARCA4*. **d–f** Representative western blots depicting global levels of the indicated proteins and histone marks (H3K27ac and H3K27me3) in the indicated cells following treatment with SAHA (**d**), in the H1299-wtSMARCA4 and H1299-mutSMARCA4 cell models after induction of SMARCA4 (dox, doxycycline, 1 μg/mL; 72 h), with or without SAHA treatment at 24 and 48 h (**e**), or in the indicated cells following treatments with SAHA and SAHA plus GSK126 (**f**). ACTIN, protein-loading control. A Source Data file is available for this figure.

known as JMJD3) have H3K27 as a substrate and play a central role in the development of some types of tumours[23]. Searching the Cancer Cell Line Encyclopedia (CCLE) we found significantly lower levels of expression of several histone demethylases (KDMs), including *KDM6A* and *KDM6B*, in SMARCA4def compared with MYCamp cells or with LC cells that are wild type for SMARCA4 and MYC (Fig. 2a and Supplementary Fig. 4a). The basal levels of EZH2 were similar in the different groups (Fig. 2b; Supplementary Fig. 4b). We validated our observations in a panel of LC cell lines at the mRNA and protein levels in which, with few exceptions, SMARCA4def cells carry lower levels of *KDM6A* and *KDM6B*, as compared to the MYCamp cells (Fig. 2b, c; Supplementary Fig. 4b). The same observations were made in lung primary tumours (Supplementary Fig. 4c). Further, the ectopic expression of the SMARCA4 wild type (H1299-wtSMARCA4 cells), triggered an upregulation of *KDM6A* and *KDM6B*, albeit subtle. An opposite effect was observed for the mutant (H1299-mutSMARCA4 cells). The levels of EZH2 were unaffected (Fig. 2d; Supplementary Fig. 4d). Conversely, in the

MYCamp cells, the depletion of SMARCA4 reduced the levels of both KDM6s (Supplementary Fig. 4e). Finally, the analysis of the changes in gene expression from publicly available databases (Sanger CRISPR) support the down-regulation of several KDMs, specially *KDM6A* and *KDM6B*, following the knockout of SMARCA4 using CRISPR/CAS9 in a panel of human cancer cell lines (https://depmap.org/portal/depmap/) (Supplementary Fig. 4f). Collectively, these observations indicate that a functional SMARCA4 is required to activate KDM6 expression.

To study the genome-wide effects of wild type and mutant SMARCA4 and of SAHA on the dynamics of H3K27 modification, we performed chromatin immunoprecipitation followed by sequencing (ChIP-seq) of SMARCA4, EZH2, H3K27ac and H3K27me3, in the H1299 cell model. No peaks were observed for SMARCA4 before adding doxycycline, which is consistent with the absence of SMARCA4 in these cells (Fig. 2e; Supplementary Fig. 5a). The global occupancy of wild type and mutant SMARCA4 was similar, indicating that ATPase activity does not influence recruitment to the chromatin (Fig. 2e–g; Supplementary

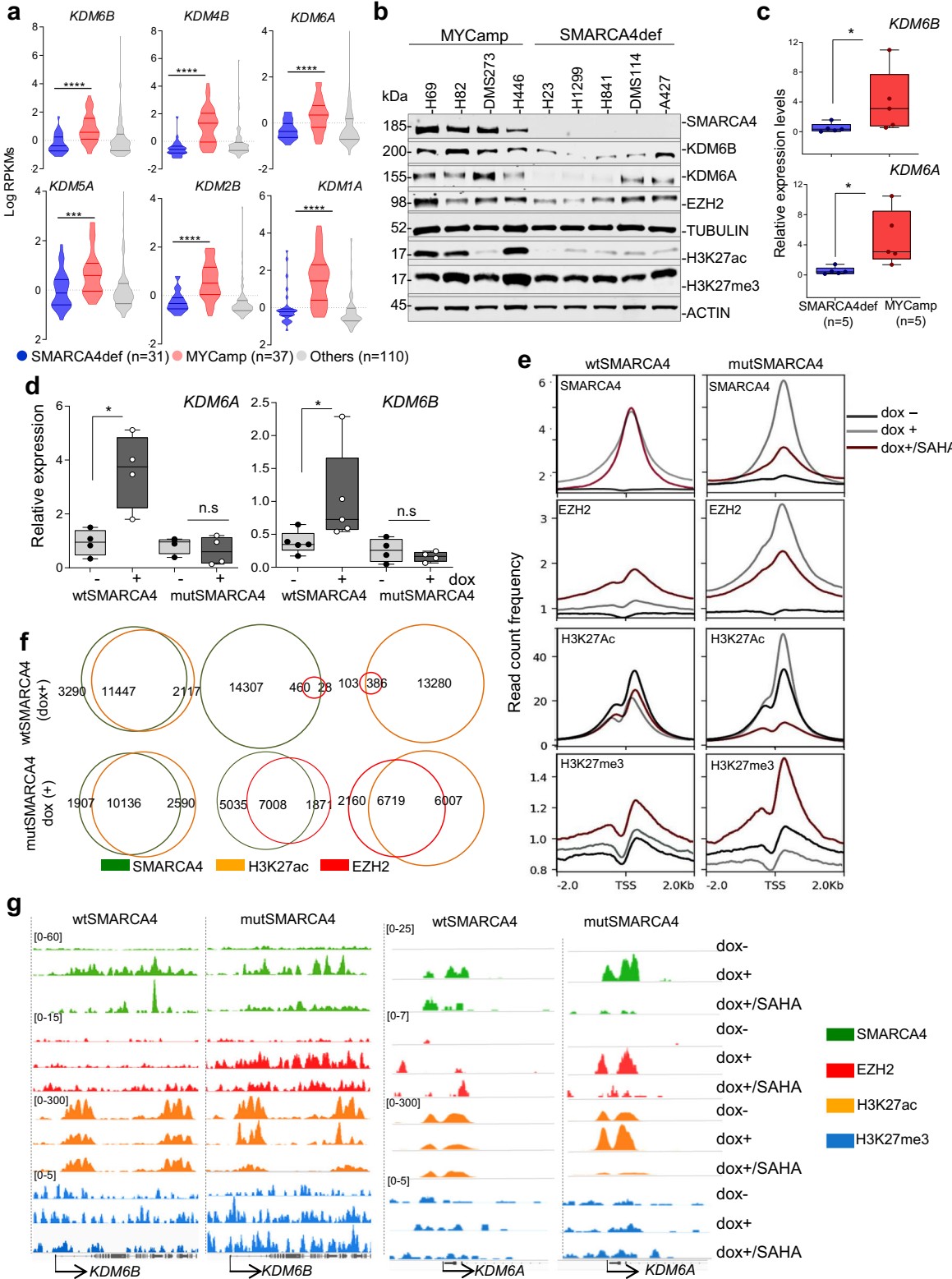

Fig. 5a, b). The H3K27ac deposition at promoters was not affected following restitution of SMARCA4 activity (Fig. 2f). However, in the SMARCA4 mutant cells, the H3K27ac peaks were sharper and it was evident a reduction of H3K27ac downstream of the peaks at promoters, in the gene body regions. This was in parallel with a strong increase in EZH2 binding to the DNA in SMARCA4 mutant-expressing cells (Fig. 2e, g; Supplementary Fig. 5c). The latter observation suggests that the overexpressed mutant protein has a dominant negative effect. H3K27ac marks were present in at least 80% of the promoters bound by SMARCA4 and some also showed EZH2 occupancy (Fig. 2f; Supplementary Fig. 6a). The increase in EZH2 binding following expression of the SMARCA4 mutant is consistent with previous findings that ectopic expression of an SMARCA4-inactive protein allows the occupancy of PRC1 and PRC2 in CpG island promoters throughout the genome[26].

**Fig. 2 SMARCA4 regulates the levels of the KDM6s. a** Violin plots comparing levels of the indicated *KDMs* in SMARCA4def ($n = 31$), MYCamp ($n = 37$) and other (wild type for *SMARCA4* and for *MYC*) ($n = 110$) lung cancer cell lines (from the Cancer Cell Line Encyclopedia—CCLE at cBioportal). RPKMs (reads per million). Bars show mean ± SD. Two-sided unpaired Student's *t*-test: ***$P < 0.001$; ****$P < 0.0001$. **b** Western blot depicting endogenous levels of the indicated proteins in lung cancer cell lines. TUBULIN and ACTIN, protein-loading control**. c** and **d** Real-time quantitative PCR of *KDM6A* and *KDM6B* (relative to *ACTB*) for comparing mRNA levels, for each individual cell line, among the indicated groups of lung cancer cell lines *$P = 0.032$ (*KDM6B*) and *$P = 0.016$ (*KDM6A*) (**c**), or in the H1299-wtSMARCA4 and H1299-mutSMARCA4 cell models (dox, doxycycline, 1 μg/mL; 72 h). *$P = 0.029$ (*KDM6A*) and $P = 0.032$ (*KDM6B*) (**d**). In box-whisker plots, the horizontal band inside box indicates the median, the bottom and top edges of the box 25th–75th percentiles and the whiskers indicate the min to max. Four and five biological replicates were included for the determination of the *KDM6A* and *KDM6B* levels, respectively, in the H1299 cell model. Bars show mean ± SD. Two-sided unpaired Student's *t*-test, **e** Read count frequency of heatmaps, at ±2 kb regions centred on the transcriptional start site (TSS), of the indicated proteins and conditions in the H1299 cell model (dox, doxycycline, 1 μg/mL and SAHA 1 μM, for 72 h). **f** Venn diagrams representing overlap of SMARCA4, H3K27ac and EZH2 peaks in the indicated cells and under the stipulated conditions. **g** Representative snapshots from IGV of ChIP-seq profiles at selected target loci performed in H1299 cell models. A Source Data file is available for this figure.

The administration of SAHA reduced the number of promoters recruiting SMARCA4 wild type by half, without producing major changes in global H3K27ac deposition (Supplementary Fig. 5a, b). SMARCA4 does not interact directly with the DNA, but rather recognises and binds acetylated lysines within histone H3 and H4 tails[27]. Given this, the severe reduction of global H3K27ac deposition observed in SMARCA4 mutant-overexpressing cells following SAHA treatment, also shown by western blot (Fig. 1e), could be the reason for the strong reduction in the global intensity of the SMARCA4 peaks (Fig. 2e). Conversely, the treatment with SAHA prompted the recruitment of EZH2 to the DNA and the increase in H3K27me3 deposition in promoter regions, in mutant and wild type SMARCA4-expressing cells. However, the effect in the latter cell type was less dramatic, and we attribute this to the slow rate of H3K27me3 removal (Fig. 2e; Supplementary Fig. 6a). The observation that SAHA increases EZH2 recruitment and H3K27me3 deposition is somewhat counterintuitive but may be a compensatory effect to avoid detrimental high levels of histone acetylation.

SMARCA4 was bound to different KDMs, including *KDM2B, KDM4B, KDM6A* and *KDM6B*, among others, in association with H3K27ac. However, there was also a concomitant increase in EZH2 occupancy in the SMARCA4 mutant-expressing cells (Fig. 2g; Supplementary Fig. 6b). These results support the idea that SMARCA4 regulates the expression of various KDMs, including *KDM6A* and *KDM6B*, through direct promoter occupancy. The increase in EZH2 in the promoter of these genes is consistent with a lack of transcriptional activation and even some transcriptional repression of these KDMs in the H1299-mutSMARCA4 cells (Fig. 2d; Supplementary Fig. 4d).

**KDM6B depletion mimics the response of SMARCA4def cells to SAHA**. We wondered to what extent the lack of KDM6A or KDM6B regulation is involved in the greater H3K27me3 and refractoriness to SAHA in the SMARCA4def cells, and whether their relative contributions differ. First, we found that the mRNA levels of the *KDM6B* were inversely correlated with the EC$_{50}$ to SAHA (Supplementary Fig. 7a). Next, using shRNAs, we down-regulated KDM6A and KDM6B expression in different MYCamp cells (Fig, 3a; Supplementary Fig. 7b), and noted that, mimicking the behaviour of the SMARCA4def cells, the reduction in *KDM6B* levels, but not of *KDM6A*, suppressed the ability to inhibit cell growth (Supplementary Fig. 7c) by SAHA. The depletion of KDM6B, but not KDM6A, also prevented SAHA from decreasing overall levels of H3K27me3 deposition (Fig. 3b), hinting at a more widespread role for KDM6B in the global removal of H3K27me3. The administration of the small molecule compound GSK-J4[28], a very specific inhibitor of KDM6A/KDM6B, has similar effects, reverting the sensitivity to SAHA in a dose-dependent manner (Fig. 3c, d) and prevented the SAHA-triggered global decrease in H3K27me3 in both KDM6s-depleted cells (Fig. 3b).

These findings suggest that a deficiency in KDM6B account for the resistance of the SMARCA4def cells to growth inhibition by SAHA.

**Inhibition of KDM6A/B is toxic in SMARCA4def cancer cells**. We hypothesised that the low levels and impaired regulation of KDM6s expression and the defects in H3K27 modification may render SMARCA4def cells particularly susceptible to KDM6s inhibition. We tested the effects of GSK-J4 on the growth of our panel of cancer cells and found that the drug was more toxic in the SMARCA4def cells, with a five-fold lower EC$_{50}$ than in the MYCamp cells or in the lung cancer cells that are wild type for both *SMARCA4* and *MYC* (Fig. 4a, b; Supplementary Fig. 8a). The greater sensitivity of the SMARCA4def cancer cells for GSK-J4 is supported by studies available from databases (Supplementary Fig. 8b). For the next stage of the study, we chose to use GSK-J4 at a concentration of 1 μM (Fig. 4a). We depleted SMARCA4 in three MYCamp cells and observed a decrease in the EC$_{50}$ for GSK-J4, which is further evidence that GSK-J4 is more toxic in cancer cells with a non-functional SMARCA4 (Fig. 4c). We also tested the effects of rescuing SMARCA4 on the response to GSK-J4 using the H1299 cell model. Overexpression of the mutant SMARCA4 increased sensitivity to GSK-J4 relative to the restitution of wild type SMARCA4 (EC$_{50}$, 0.11 μM versus 0.2 μM) (Supplementary Fig. 8c, d). The toxicity was even greater in the H1299-mutSMARCA4 than in the parental H1299 cells, supporting the existence of a dominant negative effect of over-expressing a SMARCA4-mutant protein.

GSK-J4 is a potent inhibitor of KDM6s but can also suppress the activity of other KDMs, so we investigated how the low levels of KDM6s suppress cell viability by depleting KDM6A- and KDM6B. Downregulation of KDM6A or KDM6B inhibited the growth of the SMARCA4-def cells without affecting the MYCamp cells (Fig. 4d). In a large scale public CRISPR-based genetic screen the effect of *KDM6s* knock down in cell survival of SMARCA4def cancer cell was also observed (Supplementary Fig. 8e). Moreover, the lower levels of KDM6A, and to a lesser extent of KDM6B, in the MYCamp cells, sensitised the cells to the treatment with GSK-J4 (Fig. 4e, f).

**Overexpression of KDM6A and KDM6B in SMARCA4def cells reverts sensitivity to KDM6A/B inhibition**. Next, we aimed to determine whether the increase in the levels of KDM6A and KDM6B will revert the sensitivity of the cells to KDM6A/B inhibition. To do that, we have stably overexpressed the two different KDM6s in a panel of SMARCA4def cells (H1299, H841,

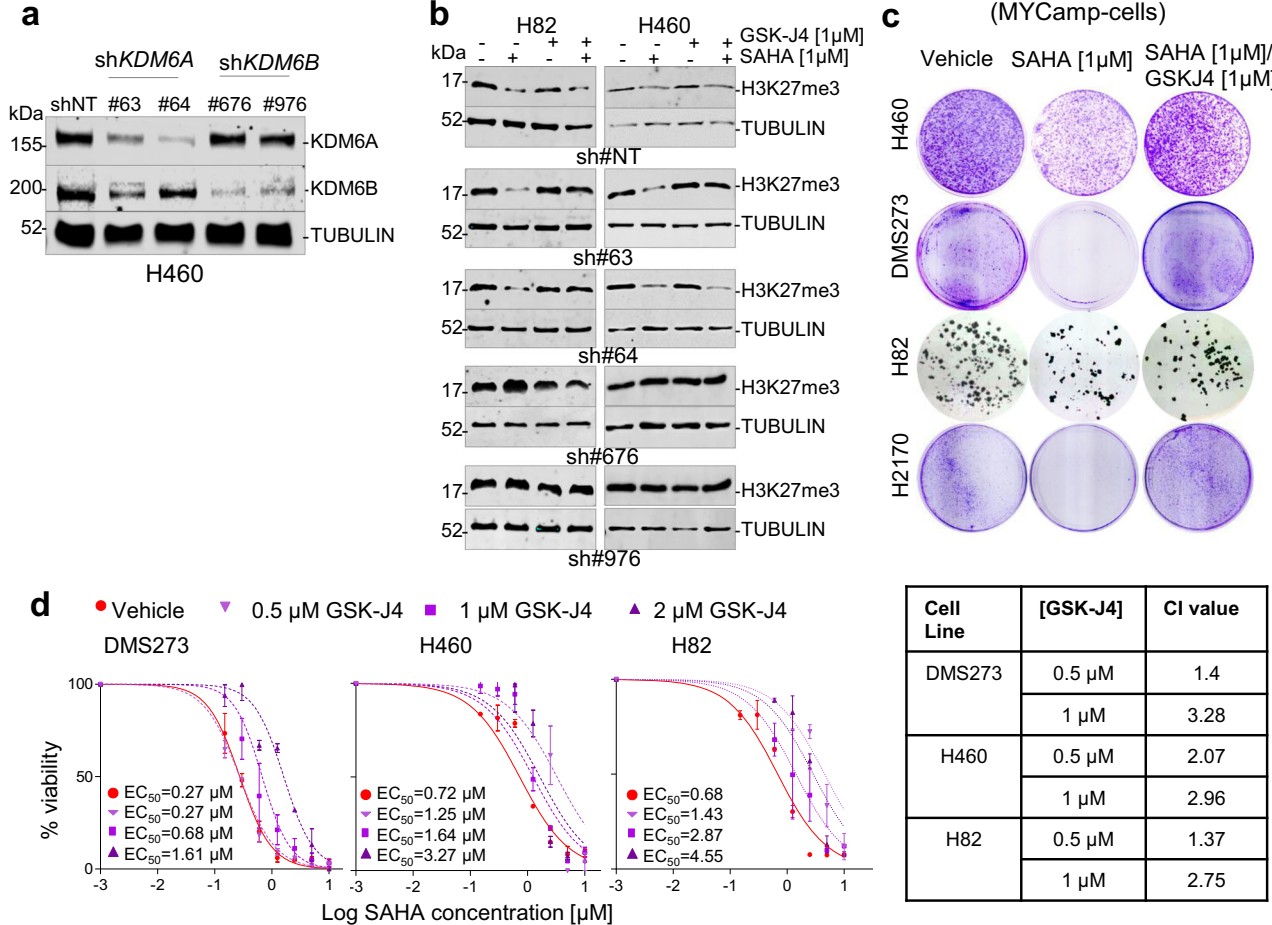

**Fig. 3 KDM6B depletion mimics the response of SMARCA4-deficient cells to SAHA. a** Western blot depicting levels of KDM6s in H460 cells infected with the non-target short hairpin (shNT) and with different shKDM6A (#63 and #64) and shKDM6B (#676, #976). **b** Western blot depicting levels of H3K27me3 and cells infected with the shNT, shKDM6A (#63, #64) or shKDM6B (#676, #976) treated with GSK-J4 and/or SAHA for 72 h. **c** Representative clonogenic assays for the indicated cells and treatments. **d** Viability of indicated cell lines, measured using MTT assays, 5 days after treatment with increasing concentrations of SAHA and co-treated or not with GSK-J4 at different concentrations. Lines, number of viable cells relative to the untreated cells. Data are presented as mean ± SD from three replicate cell cultures in two experiments. $EC_{50}$, half-maximal effective concentration. On the right, table presenting the combination index (CI) at the indicated concentrations of GSK-J4 (average CI from two independent experiments). CI < 1, CI = 1 and CI > 1 indicate synergism, additive effect, and antagonism, respectively. A Source Data file is available for this figure.

DMS114 and A427) (Fig. 5a). The overexpression of both KDM6A and KDM6B increased the $EC_{50}$ to GSK-J4 in all the cell lines tested. This effect was stronger after overexpressing the KDM6A (Fig. 5b, c).

Furthermore, we simultaneously overexpressed SMARCA4 and KDM6A or KDM6B in the H1299 cells (Fig. 5d) and measured the effects in cell viability upon treatment with GSK-J4. Wild type SMARCA4 increased the resistance to GSK-J4 mediated by KDM6A and KDM6B overexpression (Fig. 5e, f).

Together, these findings imply that the lack of SMARCA4 confers vulnerability to KDM6s inhibition on cancer cells, and that the intrinsically low levels of KDM6s, caused by the defective function of SMARCA4, underpin these effects.

**Anti-tumour effects of GSK-J4 in SMARCA4def lung cancer orthotopic mouse models.** We investigated the ability of the GSK-J4 compound to suppress tumour growth in vivo. To this end, we first grew two of the SMARCA4def (DMS114 and H841) and one MYCamp (DMS273) cell lines subcutaneously into the back of the mice (*n* = 3 mice/cell line). Once the solid tumour had entered the exponential growth phase, mice were euthanized,

and the tumours we minced into small fragments and orthotopically implanted into the lungs of another cohort of mice[20,29], to generate the orthotopic tumours. We randomly assigned the animals, implanted with each of the tumours, to treatment or vehicle groups of mice. Treatment with GSK-J4 strongly increased the overall and median survival of the animals implanted with the SMARCA4def tumours (DMS114X and H841X) relative to their matched vehicle group, whereas we found no differences between vehicle and treated groups in the animals implanted with the MYCamp tumours (DMS273X) (Fig. 6a). Remarkably, five of the mice implanted with the H841X and two of those implanted with the DMS114X and treated with GSK-J4 were alive at the end of the experiment, but had to be sacrificed despite not having respiratory difficulties or other symptoms associated with tumour progression. Our histopathological examination of tumour masses revealed the existence of large areas of necrosis in the tumours from GSK-J4-treated mice in comparison with the tumours from the vehicle-treated mice (Fig. 6b, c; Supplementary Fig. 9). We also used immunohistochemistry to determine the changes in the levels of H3K27me3 in the tumour samples following treatment with GKS-J4. We noted a significant increase in the levels of

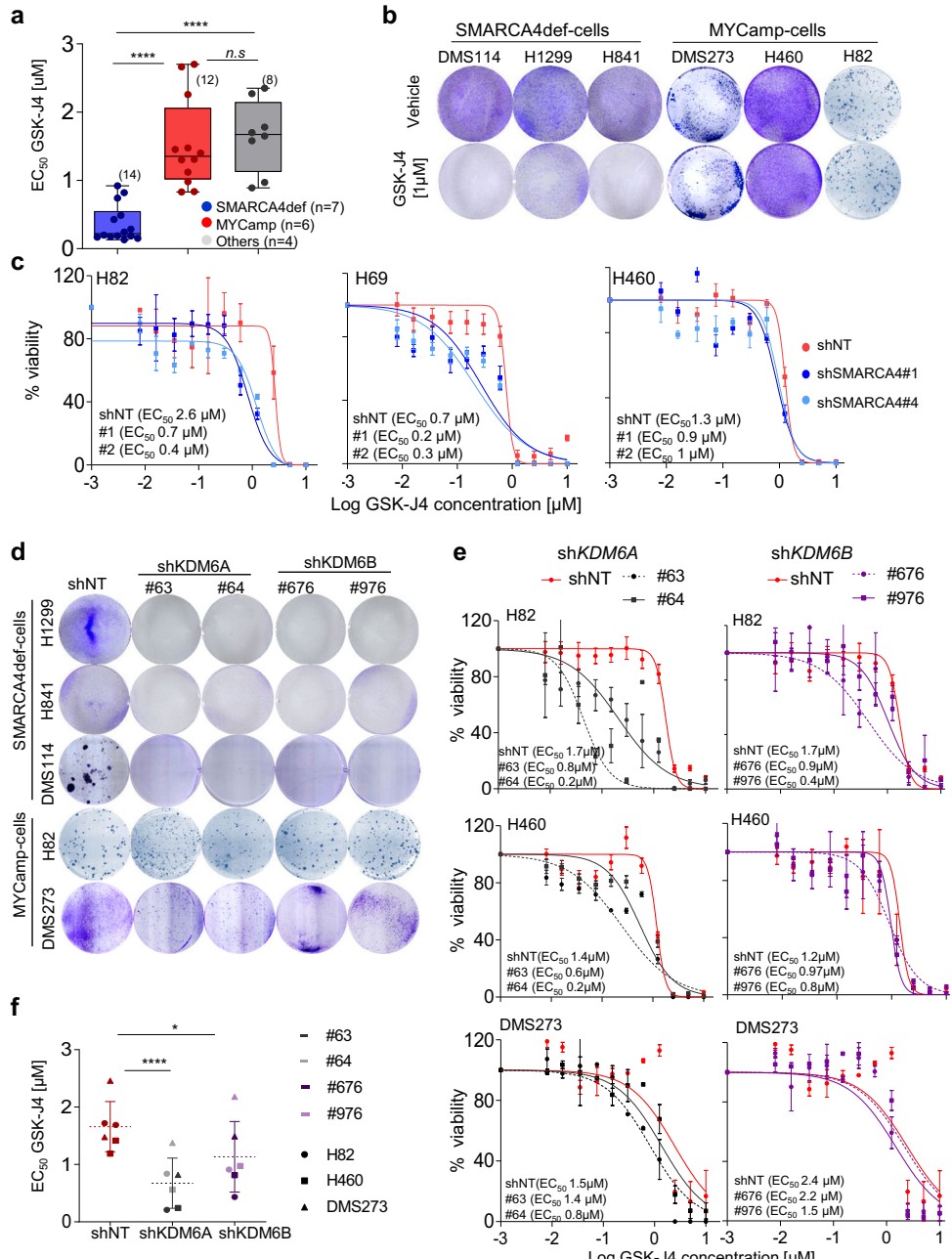

**Fig. 4 SMARCA4def cells are vulnerable to KDM6s inhibition. a** Distribution and mean of half-maximal effective concentration (EC50) values for GSK-J4 (MTT assays. Supplementary Fig. 6a) in the indicated groups of cells. Values, from each cell line and from two independent experiments are represented. Two-sided unpaired Student's *t*-test. ****$P < 0.0001$. In box-whisker plots, the horizontal band inside box indicates the median, the bottom and top edges of the box 25th–75th percentiles and the whiskers indicate the min to max. **b** Representative clonogenic assays for the indicated cells and treatments. **c** Viability of the indicated cells infected with a short hairpin non-target (shNT) control or with two shRNAs targeting SMARCA4 (#1 and #4), measured using MTT assays, after treatment with increasing concentrations of GSK-J4 for 5 days. Lines, number of viable cells relative to untreated cells. Data are presented as mean and SD from three replicates and two experiments. **d** Representative clonogenic assays for the indicated cells infected with shNT, shKDM6A (#63, #64) or shKDM6B (#676, #976). **e** Viability of the indicated cell lines, measured using MTT assays, infected with a non-target (shNT) control or with two shRNAs targeting KDM6A or KDM6B after treatment with increasing concentrations of GSK-J4 for 5 days. Lines, number of viable cells relative to the untreated cells. Error bars, mean ± SD from triplicates. **f** Distribution and mean of the EC50 from two independent experiments per each of the three LC cell lines. Lines show mean ± SD. *P*-values were calculated using paired two-tailed Student's *t* test. *$P = 0.015$; ****$P < 0.0001$; ns, not significant. A Source Data file is available for this figure.

H3K27me3 in all the tumours treated with GSK-J4, suggesting that the compound had effectively reached the tumours (Fig. 6d).

**GSK-J4 reduces tumour growth in mice implanted with SMARCA4def SCCOHT.** As previously mentioned, SCCOHT is

a very aggressive and rare type of ovarian cancer that features inactivation of SMARCA4 in almost all cases[9–11]. Here, we generated patient-derived orthotopic xenografts (PDOXs) using the primary tumours of two SCCOHT patients (OVA250 and OVA259), by orthotopically implanting the tumour in the mouse

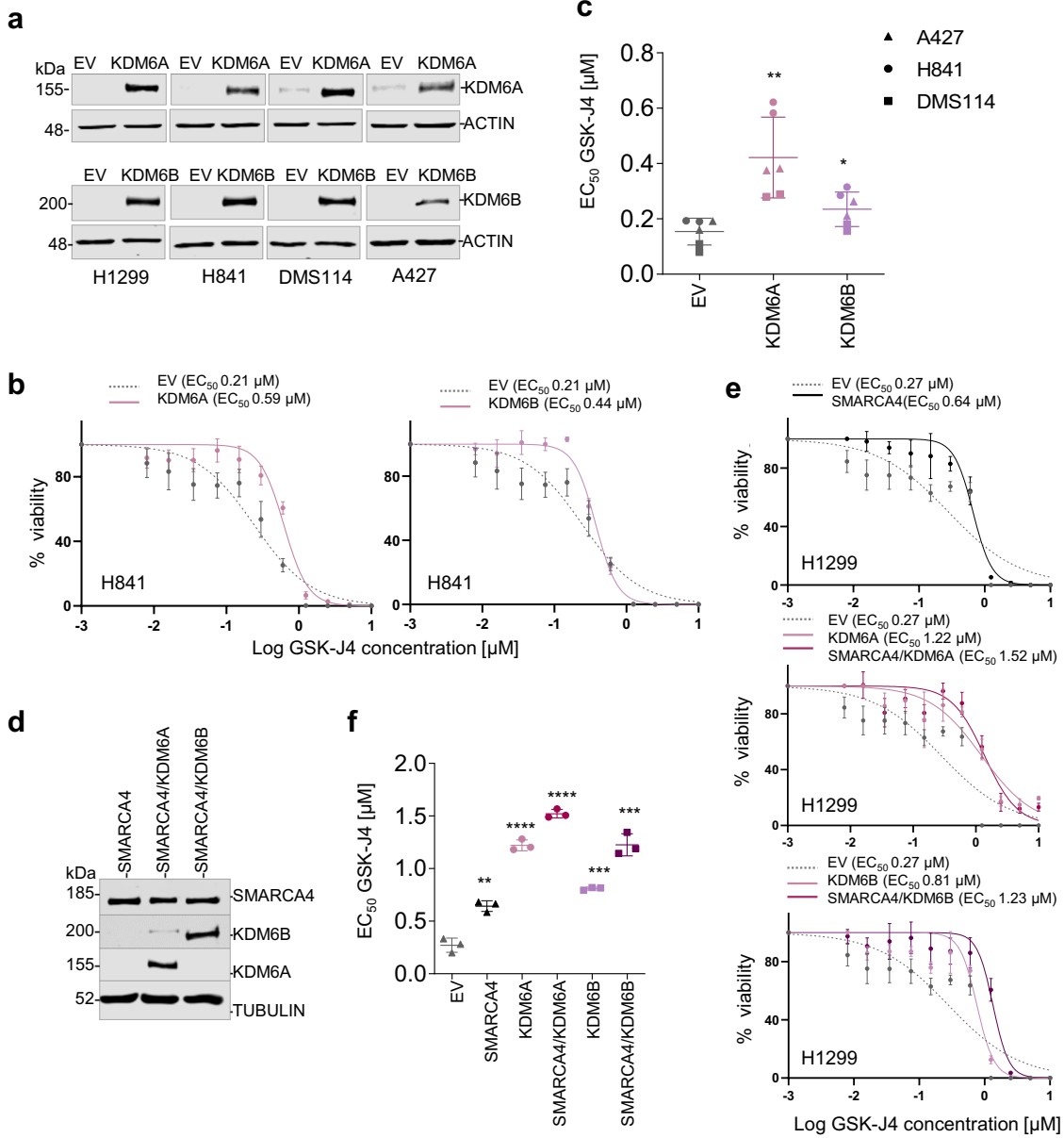

**Fig. 5 Overexpression of KDM6s reverts sensitivity to KDM6s inhibition. a** Western blot depicting levels of ectopic KDM6s and KDM6B in the indicated SMARCA4def cells overexpressing the indicated proteins. EV Empty vector. **b** Example of viability of H841 cell line, measured using MTT assays, 5 days after treatment with increasing concentrations of GSK-J4. Lines, number of viable cells relative to the untreated cells. Data are presented as mean ± SD from three replicate cell cultures in two experiments. EC$_{50}$, half-maximal effective concentration. **c** Distribution and mean of the EC$_{50}$ from two independent experiments per each of the three LC cell lines. Lines show mean ± SD. P-values were calculated using paired two-tailed Student's t test. **P = 0.0017; *P = 0.031. **d** Western blot depicting levels of KDM6s and SMARCA4 in the H1299 cells overexpressing the indicated proteins. **e** Viability of indicated cell lines, measured using MTT assays, 5 days after treatment with increasing concentrations of GSK-J4. Lines, number of viable cells relative to the untreated cells. Data are presented as mean ± SD from three replicate cell cultures in two experiments. **f** Distribution and mean of the EC$_{50}$ from three independent experiments. Lines show mean ± SD. P-values were calculated using paired two-tailed Student's t test. **P = 0.0016; ***P = 0.0002; ****P < 0.0001; A Source Data file is available for this figure.

ovary[30]. We used the PDOXs, in their first pass, to derive primary cancer cell cultures (OVA250L and OVA259L). We confirmed the presence of biallelic inactivating mutations at *SMARCA4* and the lack of protein in the two patients' tumour cells and PDOXs (Fig. 7a; Supplementary Fig. 10a). First, we tested the effects of the SAHA and GSK-J4 compounds in the primary cultures and included, as a reference, two commercial epithelial ovarian carcinoma cell lines (OVCAR-3 and OVCAR-8), which are wild type for *SMARCA4* (https://cancer.sanger.ac.uk/cell_lines). Treatment with GSK-J4 strongly suppressed cell viability and clonogenic

capability, exclusively in the OVA250L and OVA259L cells, whereas SAHA did not affect cell growth (Fig. 7b, c). The levels of induced cleavage at poly-(ADP-ribose) polymerase 1 (PARP1) were increased in the SMARCA4def cells indicating apoptosis (Supplementary Fig. 10b). It was puzzling to observe that the OVA250L and OVA259L cells had extremely low levels of global H3K27ac, even after treatment with SAHA. Similar to what happened in the lung cancer cells, the global basal levels of H3K27me3 were increased after administration of SAHA (Fig. 7d).

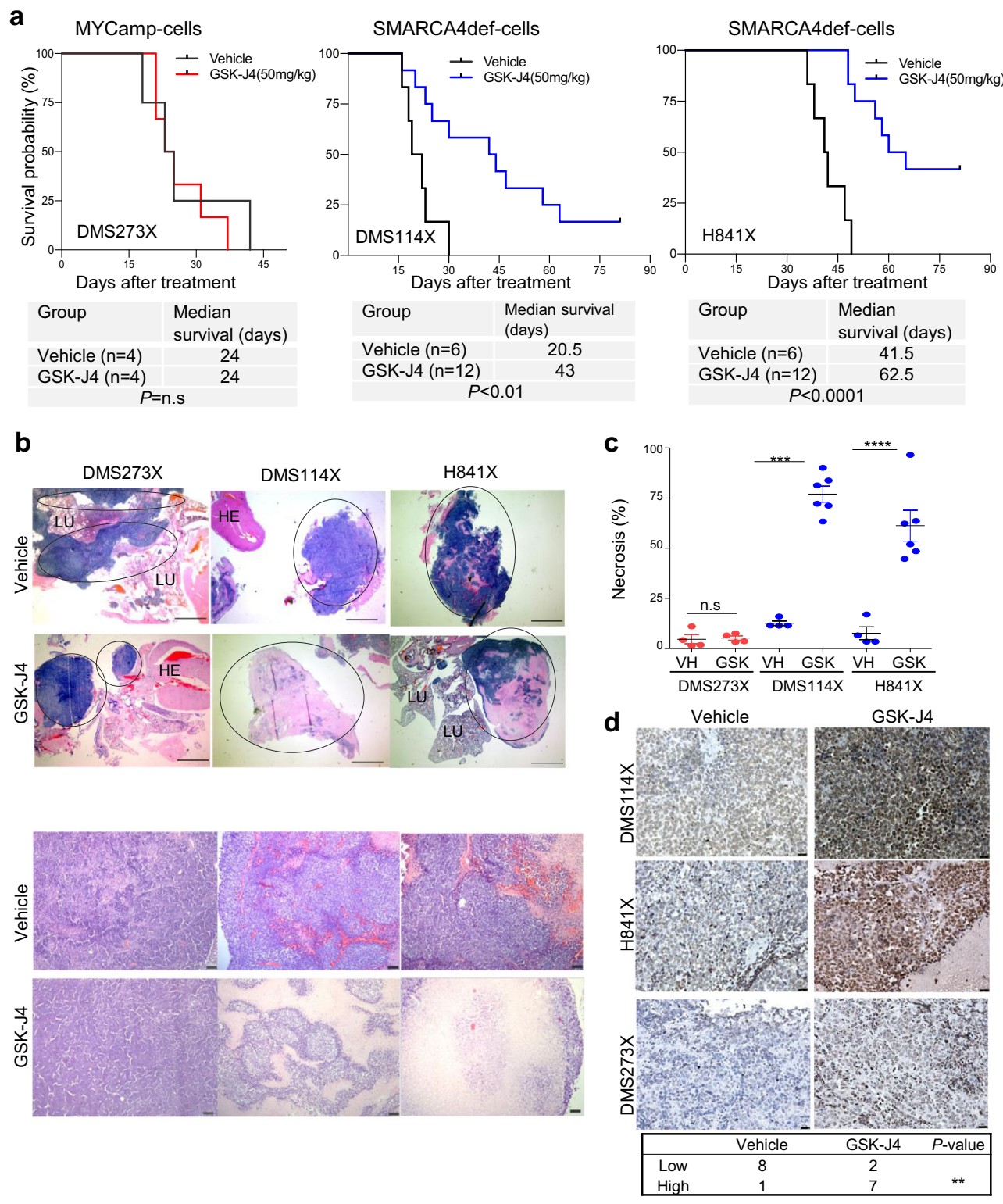

Next, we investigated the influence of GSK-J4 treatment on the growth of the OVA250 tumour in vivo. We orthotopically implanted primary tumours, either treatment-naïve or derived from mice previously treated with cisplatin (CDDP) (see the "Methods" section for further details), in the ovary of female nude mice to generate the OVA250X and OVA250XR tumours, respectively (Fig. 7e; Supplementary Fig. 10c–f). We observed a reduction in the size and weight of the tumours from mice treated with the GSK-J4 inhibitor in the OVA250X model (Fig. 7f). Although the differences did not reach statistical significance, histological examination revealed the presence of a few viable tumour cells in the tumours from mice treated with GSK-J4. These tissues contained a large amount of fibrosis, instead of the necrosis observed in the lung cancer orthotopic model, possibly

**Fig. 6 GSK-J4 induces tumour regression of SMARCA4def lung tumours in vivo. a** Kaplan–Meier curves showing overall survival for GSK-J4-treated compared with vehicle control groups of each indicated orthotopically implanted mice model. Panels below, number of mice (*n*) and mean survival times for each group of treatment and cell line. *P*-values from the two-sided log-rank (Mantel–Cox) test of the plots are included. **\*\*P < 0.01; \*\*\*\*P < 0.0001. n.s., not significant. **b** Representative sections of haematoxylin and eosin (H&E) staining of tumours from the indicated cells, from mice treated with either GSK-J4 or vehicle. Upper panels, tumour regions are marked within circles. Scale bars, 2.5 mm. The pink areas inside the tumours indicate necrosis. HE heart, LU lung. Bottom panels, representative sections, at higher magnification, of tumours from the indicated cells and treatments. Scale bars, 50 µm. **c** Quantification of necrotic areas. Mean ± SD are indicated for each group. The number of tumours, each from a different individual, were *n* = 4 in each VH and GSK-J4 treated groups for the DMS273X mice; *n* = 4 in the VH and *n* = 5 in the GSK-J4 groups, for the DMS114X mice and *n* = 4 in the VH and *n* = 6 in the GSK-J4 groups, for the H841X mice. Two-sided unpaired Student's *t*-test. **\*\*\*P = 0.0006; \*\*\*\*P < 0.0001. VH vehicle. **d** Representative immunostaining of H3K27me3 in tumours from the indicated cells and treatments. Scale bars, 25 µm. Below, distribution of H3K27me3 staining among tumours (three tumours per cell line and condition) from the DMS273X, H841X and DMS114X tumours treated with vehicle or GSK-J4. Low (intensity values 1 and 2); high (intensity values 3 and 4) (Supplementary Table 6). Two-sided Fisher's Exact test. **\*P < 0.05. A Source Data file is available for this figure.

because the OVA250X was derived from a primary tumour and not from cancer cell lines (Fig. 7g). Similar to what was observed in the lung cancer mouse models, the GSK-J4 treatment triggered an increase in H3K27me3 (Fig. 7h). Further, the levels of caspase 3, indicative of apoptosis were increased in the SMARCAdef tumours in those mice treated with GSK-J4 (Fig. 7h). Next, we generated the OVA250XR tumours to determine the benefits of GSK-J4 in tumours that have been pre-treated with CDDP, the standard treatment for SCCOHT. The morphology of the OVA250XR tumours was similar with that of the primary tumour and of the OVA250X (Supplementary Fig. 10e). The OVA250XR showed greater refractoriness to CDDP than did OVA250X (Supplementary Fig. 10f). The treatment with GSK-J4 significantly decreased tumour growth in the OVA250XR. Taken together, these observations demonstrate that the KDM6s inhibitor could constitute a therapeutic option for SCCOHT patients even after they have become resistant to CDDP (Supplementary Fig. 10g–i).

## Discussion

Here, we present evidence that cancer cells carrying oncogenic MYC are susceptible to growth inhibition by treatment with the HDAC inhibitor, SAHA. HDAC inhibitors, including SAHA, have come to be recognised as biologically active compounds of value for treating cancers, although their use is currently limited to some haematological malignancies[31]. Our current findings indicate that the pre-selection of patients with tumours in which any of the MYC family of genes have been genetically activated will have better response rates to SAHA, which suggests that SAHA could be used to treat neuroblastomas and lung cancers, among other types of cancer, in which the *MYC* genes are amplified.

Second, we found that lung or ovarian cancer cells with inactivated *SMARCA4* not only were refractory to the growth suppression triggered by SAHA, but also aberrantly accumulated H3K27me3 following the administration of this inhibitor. The levels of global H3K27ac were low in the SMARCA4def cells, a characteristic that was accentuated in the SCCOHT cells. We ruled out a central role for EZH2 methyltransferase activity in the refractoriness to SAHA in these cells, although the dependency on a non-catalytic role for EZH2 cannot be completely discounted, as it is known to affect the survival of the SWI/SNF-mutant cancer cells[14]. Additionally, we demonstrate that the transactivation of several lysine demethylases (KDMs), including KDM6A and KDM6B, is impaired in cells that lack SMARCA4, leading to a downregulation of basal KDM6s. This, coupled with inability of these cells to modulate the levels of EZH2 expression in response to SAHA and, in keeping with other knowledge[6,16–20], is evidence that defective chromatin remodelling in SMARCA4def cells promotes a closed chromatin structure and a transcriptionally rigid scenario that maintains the

refractoriness of these cells to the appropriate modification of gene expression upon different stimuli.

Despite the high degree of sequence similarity in the catalytic domain of KDM6A and KDM6B, these two enzymes have also some specific roles[32]. KDM6A is mainly associated with the demethylation of H3K27me3 at the transcriptional start sites of the HOX genes upon differentiation stimuli, whereas KDM6B is involved in inflammation and other, general physiological processes[32]. Furthermore, KDM6A, but not KDM6B, is responsible for Kabuki syndrome (KS), an infrequent, inherited disease that is characterised by neurological, endocrine and autoimmune disorders[33]. Here, we found that a deficiency in KDM6B is responsible for the refractoriness to SAHA in SMARCA4def cells. Further, a deficiency in KDM6B also accounts for the global increase in H3K27me3 upon administration of SAHA, which is consistent with a broader role for KMD6B in H3K27me3 deposition than that of KDM6A[34]. Non-catalytic activities have been proposed for the KDM6s which may also account for some of these differences[32].

Considering its potential clinical applicability, the most relevant finding presented here is the great vulnerability of the SMARCA4def cells to KDM6s inhibition, which was evident in cell culture and in mouse models with orthotopic transplants of lung cancer cells and SCCOHTs. Their frequency and poor prognosis mean that the use of GSK-J4 or similar compounds can have a great impact on the treatment of SMARCA4-mutant tumours. Likewise, SCCOHT is an aggressive carcinoma with rhabdoid characteristics and, though infrequent overall, predominantly affects young women[8–11]. Our current results, which show that GSK-J4 strongly suppresses its growth in vivo, emphasises the huge potential of KDM6s inhibitors in treating this disease, which otherwise has very limited treatment options and, consequently, a dismal prognosis. Currently, there is only limited, preclinical information about the use of KDM6s inhibitors in cancer treatment. Anti-tumorigenic activities of GSK-J4 have been shown in some leukaemias and in gliomas with *H3F3A* mutations, both of which are attributable to the inhibition of KDM6B[35,36]. In our case, the depletion of either KDM6A or KDM6B affected the viability of SMARCA4def cells, suggesting that, in this context, the depletion of either of them cannot be compensated. A limiting amount of KDM6s, due the lack of SMARCA4-mediated transactivation and regulation, seems to underlie the toxicity to the KDM6 inhibitor, GSK-J4, in SMARCA4def cells. It has also been shown that the KDM6s enhance the accessibility of the SWI/SNF complex to H3 and promote chromatin remodelling[37,38] which may also contribute to the strong dependency of SMARCA4def cells, which rely only in the SMARCA2-dependent SWI/SNF complex, on these demethylases for survival. Despite our current findings about the toxicity of the GSK-J4 compound in SMARCA4def cells, the effects could be broader, and cancer cells in which other members of the SWI/

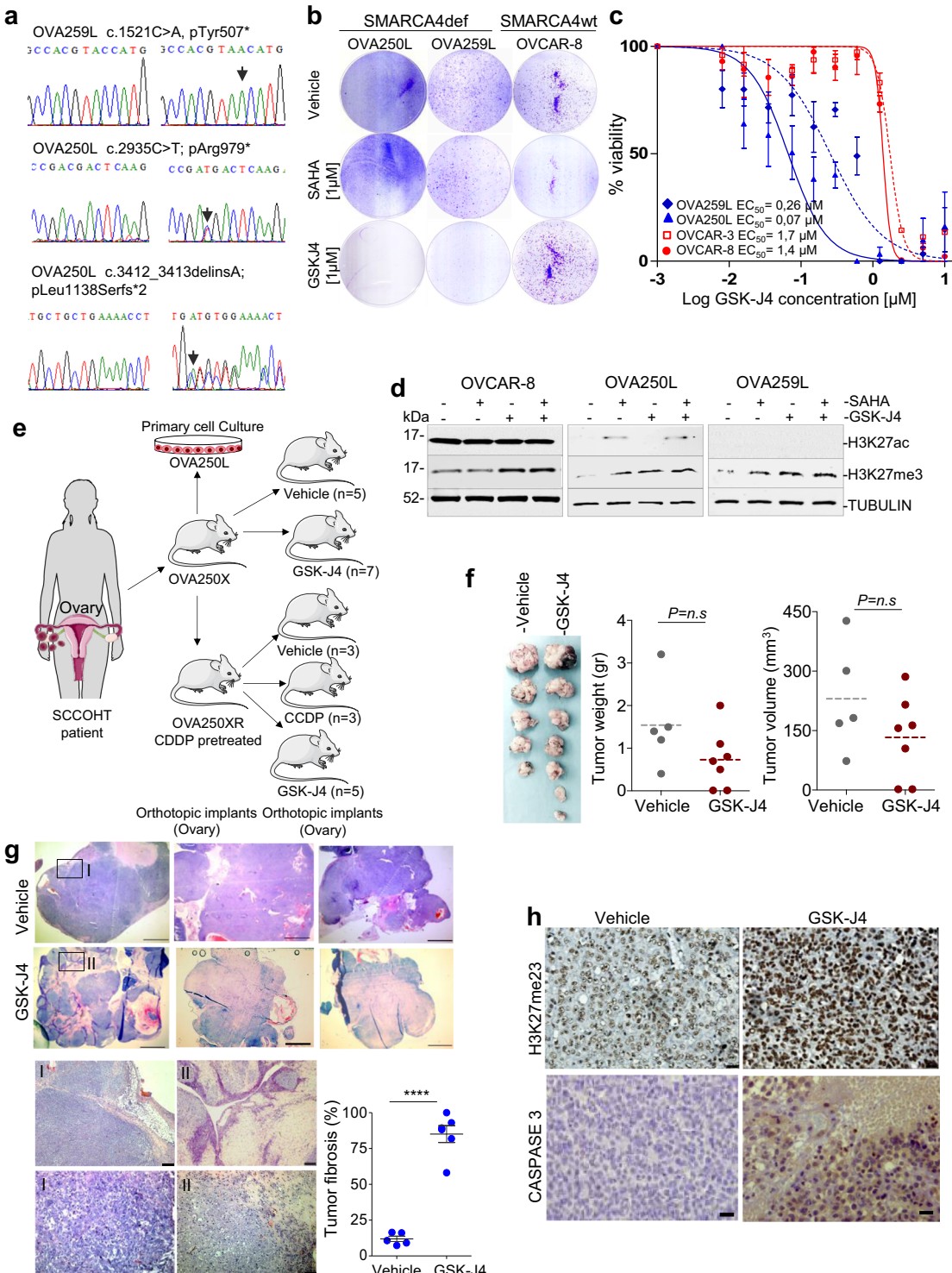

SNF-complex or of related pathways are genetically inactivated, may also be vulnerable to this inhibitor.

## Methods

**Cancer cell cultures**. The cell lines NCI-H841, NCI-H23, NCI-H82, NCI-H69, NCI-H157, NCI-H1975, NCI-H2170, NCI-H460, NCI-H1963, NCI-H446, DMS114, NCI-H727, NCI-H228, NCI-H128 and NCI-H1299 cell lines are from the American Type Culture Collection (ATCC). The DMS273 cell line is from the European Collection of Authenticated Cell Cultures (ECACC). The Lu165 cells were obtained from the RIKEN Cell Bank (Japan). The cells were grown under recommended conditions and maintained at 37 °C in a humidified atmosphere of 5% $CO_2$/95% air. The cell lines were authenticated by genotyping for *TP53* and other known mutations. All cell lines used in this study were mycoplasma-free.

Genomic DNA and total RNA were extracted by standard protocols. One primary lung cancer cell line PCD11 was derived from malignant pleural effusions[39]. Two primary cancer cell lines cultures were derived from orthoxenografts/PDOXs generated in nude mice from two primary tumours of two SCCOHT patients (OVA250L and OVA259L) that were obtained from Bellvitge Hospital and the Catalan Institute of Oncology (ICO) with the approval of the Ethical Committee (CEIC Bellvitge). Ethical and legal protection guidelines of human subjects, including informed consent, were followed. Fresh orthoxenografts/PDOXs grown in the mouse ovaries were collected when mice were sacrificed at passage #1, then minced with sterile scalpels. Single cells and clumps were transferred to cell culture plates and maintained in DMEM supplemented with 10% FBS plus 50 U/mL penicillin and 50 mg/mL streptomycin under standard culture conditions. When cell colonies with epithelial cell morphology were observed, cells were trypsinized and expanded. Both primary cell lines were considered established after >6

**Fig. 7 GSK-J4 reduces cancer cell viability of SCCOHT orthotopically implanted in mice. a** Chromatogram depicting changes in *SMARCA4*, at the genomic DNA level, in the SCCOHTs of two patients. The alterations were biallelic, confirming the complete inactivation of *SMARCA4*. A normal reference DNA is also included. **b** Representative clonogenic assays for the indicated cells and treatments. **c** Percentage of viable cells of the indicated cells, measured using MTT assays, after treatment with increasing concentrations of GSK-J4 for 5 days. Lines show the number of viable cells relative to the total number at 0 h. Error bars, mean ± SD from three replicates. EC$_{50}$ half-maximal effective concentration. **d** Western blot of the endogenous levels of the indicated proteins and cancer cells. TUBULIN, protein-loading control (48 h treatments, except 24 h in the GSK-J4 for the OVA250L cells). **e** Schematic representation of different models and treatments in mice developed from the OVA-250 tumour. SCCOHT small cell carcinoma of the ovary, hypercalcemic type. CDDP cisplatin. Figure schematics were generated using https://smart.servier.com/ (https://creativecommons.org/licenses/by/3.0/). **f** Left panel, gross pathological photographs at necropsy of the ovarian tumours that arose in mice treated with vehicle (*n* = 5) or GSK-J4 (*n* = 7). Right panels, volume and weight of each tumour. Two-sided unpaired Student's *t*-test. n.s. not significant. **g** Representative sections of H&E staining of tumours from the group of mice treated as indicated (*n* = 5 in each group). Pink areas inside the tumours indicate fibrosis. Scale bars, 2.5 mm; Lower panels, magnification of the areas indicated (rectangle); scale bar, 50 μm (above) and 25 μm (below); Lower right panel, quantification of fibrotic areas. Two-sided unpaired Student's *t*-test. Error bars, mean ± SD from replicates. ****P < 0.0001. **h** Representative immunostaining of H3K27me3 and of CASPASE-3 for OVA250X tumours treated with vehicle or GSK-J4. Scale bars, 25 μm. The quantification of the immunostaining is provided in Supplementary Table 6. A Source Data file is available for this figure.

passages in vitro. Specific informed consent was obtained from all patients for tumour implantation into mice, and the study was approved by the IDIBELL Ethics Committee (No. AAALAC-1155).

**Antibodies and western blots**. The following primary antibodies were used for western blots: anti-TUBULIN, T6199 mouse (1/10,000, Sigma Aldrich, St. Louis, MO, USA); anti-Beta-ACTIN, 13854 (1/20,000 Sigma Aldrich); anti-SMARCA4 49360S (1:1000, Cell Signaling Technology); anti-EZH2 5246S (1:1000, Cell Signaling); anti-H3K27ac D5E4 (1:1000, Cell Signaling) anti-H3K27me3 07-449 (1:1000, Cell Signaling); anti-UTX (KDM6A) D3Q1I (1:1000, Cell Signaling); anti-KDM6B (JMJD3) #3457 (1:1000, Cell Signaling) for western blots or anti-KDM6B (JMJD3) ab38113 (Abcam) for immunostaining (see also Supplementary Table 1). For western blots, whole-cell lysates were collected in a buffer containing 2% SDS 50 mM Tris–HCl (pH 7.4), 10% glycerol and protease inhibitor cocktail (Roche Applied Science). Protein concentrations were determined using a Bio-Rad DC Protein Assay kit (Life Science Research). Equal amounts of lysates (20 μg) were separated by SDS–PAGE and transferred to a nitrocellulose membrane that was blocked with 5% nonfat dry milk. Membranes were incubated with the primary antibody overnight at 4 °C, then washed before incubation with species-appropriate IRDye 680CW (925-68022) or IRDye 800CW (925-32213) fluorescent secondary antibodies (1:10,000 LI-COR, NE, USA) for 1 h at room temperature. Membrane imaging was performed on the Odyssey Li-Cor CLx i software: Image Studio Lite v5.2.

**Quantitative RT-PCR**. To assess mRNA levels of the KDMs in different cells qPCR analysis was performed. 1 μg of total RNA was reverse-transcribed using Super-Script™ II reverse transcriptase (Invitrogen) and Random primers (Promega), according to the manufacturers' instructions. qRT-PCR was performed in a Quantstudio Real-Time PCR (Quantstudio Design & Analysis v1.5.1) instrument using SYBR Green PCR Master Mix (Applied Biosystems). Three biological replicates were carried out. Primer sequences are provided in the Supplementary Table 2.

**Treatments and shRNAs**. Chemicals were obtained from the following sources: SAHA, suberoyl anilide hydroxamic acid (Cayman Chemical Company, Ann Arbour, MI, USA); GSK-J4 (Shelleckchem); GSK-126 (Cayman Chemical Company). shRNAs against SMARCA4, KDM6A and KDM6B were purchased from SIGMA-MISSION (LentiExpressTM Technology, Sigma-Aldrich) as a glycerol stock of five pLKO plasmids carrying specific shRNA sequences. A non-target shRNA (shNT) (Sigma MISSION shRNA non-mammalian control SHC002) was used as a control. The lentiviruses were generated within the 293T packaging cells. Oligonucleotide sequences are provided in the Supplementary Table 3.

**Flow cytometry**. The cells were collected with PBS and the cell suspension was transferred into the tubes, containing 70% ethanol, and fixed for 2 h. Cells were centrifuged, ethanol was decanted and cells were suspended in 5 mL PBS, and then treated with 1 mL propidium iodide (PI)/Triton X-100 staining solution (PI/Triton X-100 staining solution with RNase A, freshly made (to 10 mL of 0.1% (v/v) Triton X-100 (Sigma) in PBS add 2 mg DNase-freeRNase A (Sigma) and 200 μl of 1 mg/mL PI) for 30 min at room temperature. Flow cytometry experiments for detecting PI emission were performed in BD FACSCanto II Cell Analyzer (Becton, Dickinson) and the data analysis was performed using DNA content frequency histogram deconvolution software (FACSDiva v6.1.2). The visualisation graphs were prepared with FlowJo software version 7.6.

**Cell growth analysis and calculation of EC$_{50}$ and CI**. For cell viability assays, cell lines were incubated in 96-well plates. Before harvesting, cells were treated for

5–7 days with the indicated concentrations of each compound (SAHA, GSK-J4, GSK-126) or combinations. For the assays, 10 μL of a solution of 5 mg/mL MTT [3-(4,5)-dimethylthiazol-2-yl)−2,5-diphenyltetrazolium bromide] (Sigma Chemical Co.) was added. After incubation for 3 h at 37 °C, the medium was discarded, the formazan crystals that had formed were dissolved in 100 μL of lysis buffer (50% N-N-dimethylformamide in H$_2$O, 20% SDS, 2.5% glacial acetic acid, NaOH 5 mol/L, pH 4.7), and absorbance was measured at 596 nm. Results are presented as the median of at least two independent experiments performed in triplicate for each cell line and for each condition. For EC$_{50}$ calculations, cells were treated with each drug and their various combinations for 5 days. Estimates of EC$_{50}$ were derived from the dose response curves. To assess the drug concentration effect and to calculate the combination index (CI), cells were plated in 96-well plates and incubated with a concentration of SAHA ranging from 0.07 to 10 μM (0.07, 0.15, 0.3, 0.6, 1.25, 2.5, 5.0 and 10 μM), and the same for GSK-J4 for 5 days. MTT assays were performed and the EC$_{50}$ was determined by fitting the dose-response curve utilising the CompuSyn software. The CI values for each dose and the corresponding effect level were calculated. The CI offers a quantitative definition for drug combinations in which CI < 1, CI = 1 and CI > 1 indicate synergism, an additive effect, and antagonism, respectively.

For clonogenic assays, the plates were seeded with 5000 cells from each cell line, then treated with SAHA (1 μM), GSK-J4 (1 μM) or GSK-126 (1 μM) for 5 days. Cells were stained with crystal violet solution (0.5% Crystal Violet in 25% of methanol).

**ChIP-sequencing**. For ChIP, cells were grown in P-150 cm cell dishes and fixed with 1% methanol-free formaldehyde (Thermo Scientific) for 10 min at room temperature, then quenched by 125 mmol/L glycine for 15 min at room temperature, washed with ice-cold PBS twice and centrifuged at 200 × *g*, 4 °C for 5 min. The pellet was resuspended in 1 mL of cell lysis buffer (10 mmol/L Tris–HCl, 10 mmol/L NaCl, 0.5% NP-40, protease inhibitor) and kept at 4 °C, rotating for 30 min. After centrifugation, the pellet was resuspended in 1 mL of nuclear lysis buffer (1% SDS, 10 mmol/L EDTA, 50 mmol/L Tris-HCl pH 8.0, protease inhibitor) and kept at 4 °C for 60 min. After another centrifugation, the lysate was sonicated with a Covaris M220 instrument to yield chromatin fragments of an average size of 0.25–1.00 kb, and then frozen at −20 °C for 30 min. The chromatin was thawed on ice and centrifuged at 2500 × *g*. For each ChIP reaction, 60 μL of Magna ChIP™ Protein A + G Magnetic Beads (Merck Millipore) was used in accordance with the manufacturer's protocol. Before addition of the sheared chromatin to the beads, Triton X-100 and Na-deoxycholate was added to a final concentration of 10% each. 1% of the chromatin volume was used for input. At least two independent ChIP experiments were performed.

Immunoprecipitated chromatin was deep-sequenced in the Genomics Unit of the Centre for Genomic Regulation (CRG, Barcelona, Spain) using the Illumina HiSeq 2500 system and HiSeq Control Software (HCS) 2.2.68 software (Illumina). Briefly, library preparation included end-repair, generation of dA overhangs, adapter ligation, size selection and removal of non-ligated adapters by agarose gene electrophoresis and amplification (18 cycles) before loading the samples into the sequencer.

For ChIP-sequencing data analysis, reads were aligned to the human reference genome hg38, using Bowtie v1.2.2, with default parameters and disallowing multi-mapping (−m 1)[40]. PCR duplicates were removed using PICARD (http://broadinstitute.github.io/picard/). Ambiguous and multi-mapped reads were discarded. Peaks were called using MACS2 v2.1.1[41]. To avoid false positives, peaks were discarded if they were present in the ChIP-seq of SMARCA4 in the SMARCA4-deficient cells. Genomic peak annotation was performed with the R package ChIPpeakAnno v3.15, considering the region ±2 kb around the TSS as the promoter[42]. All analyses considered peaks overlapping with promoter regions, unless otherwise specified. Peak lists were then transformed to gene target lists.

For heatmap and intensity plot representation of ChIP-seq signal, bedgraph files were generated using the makeUCSCfile function in HOMER with default parameters normalising for differences in sample library size, and BigWig files were generated using the function bedGraphToBigWig from UCSC. Heatmaps were derived using the functions computeMatrix, in a window of ±2 kb around the centre in the TSS, and plotHeatmap from deepTools[43] (version 3.5).

**Construction of expression vectors and infections**. The complete KDM6A cDNA (NM_021140.4) was PCR-amplified, from a retrotranscribed human RNA pool (Agilent Technologies, Santa Clara, CA, USA) using Phusion High-Fidelity DNA Polymerase (Thermo Scientific, Waltham, MA, USA) following standard protocols. The cDNA fragments were cloned into the pLVX-IRES-ZSGreen vector (Clontech), between the XhoI and NotI endonuclease restriction sites, following the manufacturer's instructions. For the KDM6B, the complete KDM6B coding region (ENSG00000132510.10) was PCR-amplified, from a human DNA pool (Agilent Technologies, Santa Clara, CA, USA) using Phusion High-Fidelity DNA Polymerase (Thermo Scientific, Waltham, MA, USA), following standard protocols. The fragment was cloned into the pCDNA4/TO vector (Invitrogen), between the KpnI and XhoI endonuclease restriction sites, following the manufacturer's instructions. The KDM6A and KDM6B inserts were then Sanger-sequenced. All the primers used are listed in Supplementary Tables 4 and 5. For KDM6A ectopic expression, the cells were infected with lentiviruses derived from the pLVX-IRES-ZSGreen vector and then selected by sorting GFP positive cells, using an BD FACSArea (Becton, Dickinson). For KDM6B ectopic expression, the cells of interest were transfected with a pCDNA4/TO-KDM6B construct and clones expressing KDM6B were selected with Zeocin (InvivoGen) and pooled.

**Generation of orthotopic tumour models and treatments**. Male and female athymic nu/nu mice (ENVIGO) aged 4–5 weeks were maintained in a sterile environment before use in the lung cancer orthotopic experiments. Female athymic nu/nu mice (ENVIGO), 4–6 weeks old, were used for the ovarian cancer orthotopic studies. The animals were housed in individually ventilated cages on a 12-h light–dark cycle at 21–23 °C and 40–60% humidity. Mice were allowed free access to an irradiated diet and sterilised water. All animal experiments were approved by the IDIBELL Ethical Committee under protocol 9111 approved by the Government of Catalonian, AAALAC accredited Unit 1155, and performed in accordance with guidelines stated in the International Guiding Principles for Biomedical Research Involving Animals, developed by the Council for International Organizations of Medical Sciences (CIOMS). To generate orthotopic lung tumour xenografts the cell lines were injected subcutaneously into the back of the mice ($n = 3$ mice/cell line). Once the solid tumour had entered the exponential growth phase, mice were sacrificed, the tumour was isolated under sterile conditions, and the non-necrotic areas were selected and minced in small fragments of 2–3 mm$^3$. These were then orthotopically implanted in the lung parenchyma[20,29]. On day 15, the mice were randomised and intraperitoneally treated with GSK-J4 (50 mg/kg/day for each mouse) or corresponding vehicle only. For the lung orthotopic models, in most cases the animals were sacrificed when they displayed serious respiratory difficulty, which was subsequently confirmed to be associated with lung tumour growth.

Orthoxenografts or PDOXs of SCCOHT were generated. The primary tumour specimens for the two primary SCCOHT samples were freshly obtained at Hospital Universitario de Bellvitge (Hospitalet de Llobregat, Barcelona, Spain). The study was approved by the Institutional Review Board, and written informed consent was obtained from both patients. The orthotopic ovarian tumours were engrafted in mice, following an established protocol[30]. Briefly, non-necrotic tissue pieces (2–3 mm$^3$) from resected carcinoma were selected and placed in DMEM (BioWhittaker) supplemented with 10% FBS and penicillin/streptomycin at room temperature. Under isofluorane-induced anaesthesia, animals were subjected to a lateral laparotomy, their ovaries exposed, and tumour pieces anchored to the ovary surface with prolene 7.0 sutures. Tumour growth was monitored 2 or 3 times per week and when the tumour had reached a sufficient size, it was harvested, cut into small fragments, and transplanted into between two and four new animals. Engrafted tumours (named OVA250X) at early mouse passages were cut into 6–8 mm$^3$ pieces and stored in liquid nitrogen in a cryopreservation solution of 90% FBS and 10% dimethyl sulfoxide, awaiting subsequent implantation.

To generate the cisplatin-resistant ovarian xenograft mouse model, orthotopically engrafted OVA250X tumours at passage#1 were allowed to grow ($n = 3$ mice) until palpable intra-abdominal masses were noted. Cisplatin was i.v.-administered to the animals (cycle #1, 3, 5 mg/kg dose) for 3 consecutive weeks (days 0, 7 and 15; cycle#1 of treatment) (Supplementary Fig. 9b). Post-cisplatin tumours at relapse were harvested and engrafted in new animals. This process of cisplatin treatment was repeated up to four times by treating tumour-bearing mice with stepwise-incremental doses of cisplatin: cycle #2, 4 mg/kg; cycle #3, 5 mg/kg and cycle #4, 5 mg/kg (see Supplementary Fig. 9b). Cisplatin-resistant tumours were obtained (OVA250XR). At doses higher than 3.5 mg/kg, signs of cisplatin induced some toxicity that were ameliorated by 2 days administration of saline containing 5% glucose. Mice were transplanted with fragments of OVA250X and OVA250XR tumours, and when tumours reached a homogeneous palpable size were randomly allocated into the treatment groups ($n = 3$-7 mice/group): (i) Placebo; (ii) GSK-J4 (50 mg/kg) and (iii) cisplatin (3.5 mg/kg); drugs were administered once a day, 5 days per week, for 4 consecutive weeks. Animals were sacrificed on day 21 of treatment, and their ovaries dissected out and weighed. Representative fragments were either frozen in nitrogen or fixed and processed for paraffin embedding.

**Histopathology and immunostaining**. For histological analysis, tumours were fixed and embedded in paraffin. Necrosis/fibrosis were morphological assessed after staining with haematoxylin and eosin (H&E), using standard protocols, and then examined by light microscopy in a blinded fashion. For immunostainings, 4-μm-thick paraffin-embedded sections of lung and ovarian tumour samples were deparaffinized overnight at 62 °C and then immersed in xylene. Samples were rehydrated and, after microwaving with Tris/EDTA pH 9.0 for antigen retrieval, endogenous peroxidase was inhibited with a 3% hydrogen peroxide solution, blocked in 10% goat serum and incubated with primary antibodies overnight at 4 °C (Supplementary Table 1). HRP-conjugated polyclonal goat (anti-mouse and anti-rabbit) secondary antibodies (NeoStain ABC Kit, NeoBiotech) were used in 1-h incubations at room temperature. Labelling detection was done using an ImmPACT DAB Peroxidase (HRP) Substrate kit (Vector Laboratories, Burlingame, CA, USA), and tissue sections were counterstained with haematoxylin. Once dehydrated in an ethanol battery and cleared in xylene for 1 h, samples were mounted with coverslips with DPX mounting medium (Merck Millipore, Darmstadt, Germany). Sections were evaluated under a Leica DM1000 microscope by two independent observers in a blind fashion. Areas of necrosis/fibrosis were quantified using Photoshop (version 2021). The scoring criteria for determining H3K27me3 staining were based on the staining intensity (four categories, 1–4) (Supplementary Table 6). The mean of values from three independent evaluators was determined.

**Statistical analysis**. Student's $t$-tests, EC$_{50}$ calculations, Kaplan–Meier estimates and log-rank (Mantel–Cox) test were performed using Prism software (GraphPad Prism 9). Values of $P < 5\%$ were considered statistically significant. The statistical methods used for each analysis are specified in the figure legends.

**Reporting summary**. Further information on research design is available in the Nature Research Reporting Summary linked to this article.

## Data availability

The ChIP-seq raw data obtained in this study has been uploaded to the Gene Expression Omnibus-GEO (NCBI), under accession number GSE155129. Uncropped western blot images are also present in a Source Data file. Databases/Datasets used in the study were: cBioPortal for Cancer Genomics, Cancer Cell Line Encyclopedia-CCLE (https://www.cbioportal.org/study/summary?id=ccle_broad_2019), COSMIC-Catalogue of Somatic Mutations in Cancer (https://cancer.sanger.ac.uk/cell_lines), DepMap Portal-Cancer Dependency Map (https://depmap.org/portal/achilles/), Genomics of Drug Sensitivity in Cancer database (https://www.cancerrxgene.org/compound/Vorinostat/1012/overview/ic50, https://depmap.org/portal/GSK-J4?tab=dependency). The authors declare that other data supporting the findings of this study are provided in the Supplementary Information/Source Data file. Source data are provided with this paper.

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

## Acknowledgements

The authors thank Isabel Bartolessis (Cancer Genetics Group) at IJC for technical assistance. This work was supported by the Spanish Ministry of Economy and Competitivity-MINECO (grant number SAF-2017-82186R, to M.S.-C., and grant PI19/01320 to A. Villanueva) and from the Fundación Científica of the Asociación Española Contra el Cancer (AECC) (grant number GCB14142170MONT) to M.S.-C. A. Villanueva is also funded by the Department of Health of the Generalitat de Catalunya (2014SGR364). O.A. R. received a Juan de la Cierva postdoctoral contract (grant No. IJCI-2016-28201, until November 2019) and an AECC research contract (INVES19045ROME from December 2019). A. Vilarrubi, P.L. and A.A. are supported by pre-doctoral contracts from the Spanish MINECO (FPI-fellowship: PRE2018-084624, BES-2015-072204 and FPU17/00067). M.S. was supported by a Rio Hortega contract from the Instituto de Salud Carlos III (CM17/00180). L.F. received a European Union Horizon 2020 research and innovation programme under the Marie Sklodowska-Curie Actions grant agreement, number 799850.

## Author contributions

O.A.R. and M.S.-C. conceived and designed the study; An.V., Al.V., A.G., A.A., D.T., E.P., J.J.A.-B., F.S., P.L. and S.V. performed the experiments and analysed data with assistance from L.F., J.F.M.-T., M.S.; A.O., J.M.P., X.M.-G., P.P.M. and Au.V. provided essential reagents and intellectual input. O.A.R. and M.S.-C. coordinated the project, interpreted results, wrote the manuscript and supervised the project.

## Competing interests

Al.V. and Au.V. are co-founders of Xenopat S.L. The remaining authors declare no competing interests.
