## [Peer Review File · Nature Communications]

REVIEWER COMMENTS

Reviewer #1 (Remarks to the Author):

In this report, the authors provide evidence that SMARCA4 deficient tumors are hypersensitive to KDM6A and B suppression/inhibition. This reviewer provided feedback on the original manuscript and was asked to assess the revised manuscript, which was transferred to Nature Communications from Nature Genetics.

The authors have mostly addressed my questions. However, it is requested that the authors rescue the sensitivity to KDM6/B knockdown and pharmacological inhibition by reconstitution of KDM6A/B expression in SMARCA4 deficient cancer cells. This is required because of the paradoxical result in that in most cases dependence on a particular cancer-relevant gene (here, KDM6A and B) occurs in the context of gene upregulation instead of downregulation. This paradoxical finding should also be discussed more completely in the text. This experiment will also help confirm the specificity of the inhibitor effects.

Reviewer #3, Replacement Reviewer for Reviewer #2 (Remarks to the Author):

Comments on authors' response to prior review.

1. Addressed satisfactorily. Please replace Fig. 1d with the new data.
2. The explanation provided here did not address the question "why the H1299-mutSMARCA4 cells underwent an increase in H3K27me3 concomitantly with a decrease in global H3K27ac" as the explanation addressed changes in distribution but not changes in global H3K27ac. Fig. 2e, the explanation provided with regard to SMARCA4 association is mediated by H3K27ac was not supported by the data in Fig. S3a (SAHA did not affect H3K27ac differentially). Fig. 2g, it's unclear what the authors try to convey here. Based on the data, SAHA should increase KDM6A/6B in muSMARCA4 expressing cells, which would be contrary to the authors' model. The authors may discuss this.
3. Addressed satisfactorily, but there is not Supplementary Fig. 3d and 3e in the supplementary figures that match the authors' description.
4. Addressed satisfactorily.
5. Addressed, but the figure was not replaced in Fig. 2b. However, the data as presented do not support the conclusion that "the low levels of KDM6s, basal global H3K27ac was present at a lower level in the SMARCA4def cells than in the MYCamp cells" For example, DMS273 is high in KDM6A, but low in H3K27ac. In contrast, H841 is high in KDM6B (perhaps the highest). The authors' statement on KDM6A/6B and H3K27ac is also not relevant as KDM6A/6B do not directly regulate H3K27ac, and there is no correlation between KDM6A/B and H3K27me3 (that KDM6A/6B do regulate).
6. Not addressed satisfactorily. Supplementary Fig. 2c (not supplementary Fig. 2b), there was a clear decrease in both KDM6A/6B by westernblot. In contrast, the increase by wtSMARCA4 for KDM6A is really marginal at the best. The authors will need to address this.
7. Not addressed satisfactorily. The westernblot (now insupplementary Fig. 2d) clearly showed that in DMS273 cells SMARCA4 knockdown by #4 shRNA did not decrease KDM6B and both shSMARCA4s did not decrease KDM6A. Again, the authors will have to address why there is a discrepancy between their westernblot and qRT-PCR and why the authors choose to believe the results of qRT-PCR over westernblot.
8. Addressed satisfactorily.
9. Addressed satisfactorily.
10. Addressed satisfactorily.
11. The explanation is acceptable.

12. Addressed satisfactorily.

13. Acceptable, but would be better to correlate the IC50 changes with the pathway characterized here and in particular for those MYC normal and SMARCA4 def vs. proficient cells (others as labeled) (SMARCA4, KDM6A/B, H3K27ac and H3K27me3, EZH2 etc in this panel of cell lines).

14. This reviewer agrees with the prior reviewer that the difference in marginal and in particular in DMS273 cells and shKDM6B in H460 cells. The conclusion is an over interpretation of the data presented.

15. Addressed satisfactorily.

This reviewer did not review the original submission and conducted an independent review prior to receiving the authors' response. As such, to be fair to the authors, the following points are new. However, this reviewer believes these control experiments should be included to limit off target effects.

Major comments:

1. The rationale for targeting KDM6A/6B in KDM6A/6B low SMARCA4 inactivated cells is missing. What is the mechanism that the authors propose to account for these findings? At the minimum, inducible knockdown of KDM6A/6B in isogenic SMARCA4 proficient vs. deficient cells should be used as a control for on target effects. Conversely, if the model as proposed by the authors are correct, ectopic expression of KDM6A/6B in SMARCA4 deficient cells should rescue the growth inhibition induced by GSK-J4.

2. The findings related to HDAC inhibition and EZH2 inhibition are contrary to a number of publications in the literature (e.g., Wang et al, J Path, 2017; Chan-Penembre et al, Mol Cancer Ther, 2017, Wang et al, Mol Cancer Ther, 2018). The authors completely ignored these prior findings and did not provide any explanation.

3. There is no control for off-target effects of GSK-J4 in their experimental design and this is important given GSK-J4 suppresses a number of KDMs. For example, it would be informative to test GSK-J4 in KDM6A/B knockout cells.

4. Given the differential responses to SAHA, the changes in various factors/markers could simply be due to changes in cell growth kinetics/cell cycle distribution that are known to regulate relevant activities such as EZH2. Fig. 1: d, cell cycle distribution should be examined. e, there is no decrease in H3K27ac in wtSMARCA4 re-expressed cells, indicating off-target effects of SAHA. f, why SAHA decreases EZH2 and GSK126 further decreases EZH2? Fig. S1b, why and how did SAHA change (decrease or increase) EZH2 expression? Cell cycle distribution should be carefully examined.

5. The authors observed upregulation of H3K27me3 by SAHA in SMARCA4 deficient cells, and hypothesize that this might be the reason of resistance to SAHA. However, restoration of H3K27me3 by GSK-J4 failed to re-sensitize SMARCA4 inactivated cells to SAHA, which argues against this model. In addition, GSK126 obviously caused resistance to SAHA in DM273 cells, while the authors claimed GSK126 did not affect sensitivity and EZH2 is not implicated. At least, it is possible that EZH2 participates the repression of KDM6A/B caused by mutSMARCA4.

6. Based on Fig. 2e and g, increased binding of EZH2 and H3K27me3 by mutSMARCA4 in the KDM6B promoter could be the reason why KDM6A/B expression was decreased in mutSMARCA4 cells. Did the authors check the role of EZH2 in mutSMARCA4 mediated KDM6A/B decrease? In addition, why are KDM6A/B upregulated by wtSMARCA4? Why was SAHA not effective in decreasing H3K27ac in wtSMARCA4 in Fig. 1e and Fig.S3a.

Minor:

1. Fig. 5d, DMS273X tumors should be included as controls.

2. Fig. 6: b, the quality of the image is poor. d, Why GSJ-J4 did not increase H3K27me3? f, the tumor volume does not appear to be consistent with the image. h, data should be quantified.

3. Fig. S5: the data as presented would only support KDM6A, but not KDM6B.

4. Fig. S6: the differences in panel d (in the figure) are marginal at the best.

5. The authors should clearly label the cell lines as MYCamp or SMARCA4 mutant in every figure.

6. There are many data as described in the text were not presented and in particular those related to database mining (in addition to mis-labeling). They include S3c, S2d-e, S6b, S6e etc.

REVIEWERS' COMMENTS:

SECOND REVIEW PROCESS

Reviewer #1 (Remarks to the Author):

In this report, the authors provide evidence that SMARCA4 deficient tumors are hypersensitive to KDM6A and B suppression/inhibition. This reviewer provided feedback on the original manuscript and was asked to assess the revised manuscript, which was transferred to Nature Communications from Nature Genetics.

The authors have mostly addressed my questions. However, it is requested that the authors rescue the sensitivity to KDM6/B knockdown and pharmacological inhibition by reconstitution of KDM6A/B expression in SMARCA4 deficient cancer cells. This is required because of the paradoxical result in that in most cases dependence on a particular cancer-relevant gene (here, KDM6A and B) occurs in the context of gene upregulation instead of downregulation. This paradoxical finding should also be discussed more completely in the text. This experiment will also help confirm the specificity of the inhibitor effects.

We acknowledge the importance of overexpressing KDM6A/B proteins in SMARCA4def cancer cells, to ascertain whether this can revert the sensitivity of the cells to KDM6A/B inhibition. To do that, we now have used different lung cancer SMARCA4def cells, the H1299, H841, DMS114 and A427 cells and have overexpressed the two different KDM6s. In the H1299 cells, we have also simultaneously restored the expression of SMARCA4wt. We believe that these data support our conclusions that low levels of KDM6s underlie the sensitivity of SMARCA4def cells to KDM6 inhibition. This new data has been included in an additional sub-section of the results (on page 11). It reads as follows:

Overexpression of KDM6A and KDM6B in SMARCA4def cells reverts sensitivity to KDM6A/B inhibition. Next, we aimed to determine whether the increase in the levels of KDM6A and KDM6B will revert the sensitivity of the cells to KDM6A/B inhibition. To do that, we have stably overexpressed the two different KDM6s in a panel of SMARCA4def cells (H1299, H841, DMS114 and A427) (Fig.5a). The overexpression of both KDM6A and KDM6B increased the EC50 to GSK-J4 in all the cell lines tested. This effect was stronger in the KDM6A overexpressing cells (Fig.5b-c).

Furthermore, we simultaneously overexpressed SMARCA4 and KDM6A or KDM6B in the H1299 cells (Fig. 5d) and measured the effects in cell viability upon treatment with GSK-J4. Wild type SMARCA4 increased the resistance to GSK-J4 mediated by KDM6A and KDM6B overexpression (Fig. 5e-f).

Together, these findings imply that the lack of SMARCA4 confers vulnerability to KDM6s inhibition on cancer cells, and that the intrinsically low levels of KDM6s, caused by the defective function of SMARCA4, underpin these effects.

This new information also includes an additional figure (Figure 5) (the figures that follow have been renamed accordingly), another sub-section in the methods and the following sentence in the discussion.

“A limiting amount of KDM6s, due the lack of SMARCA4-mediated transactivation and regulation, seems to underlie the toxicity to the KDM6 inhibitor, GSK-J4, in SMARCA4def cells. It has also been shown that the KDM6s enhance the accessibility of the SWI/SNF complex to H3 and promote chromatin remodelling³⁷⁻³⁸ which may also contribute to the strong dependency of SMARCA4def cells, which rely only in the SMARCA2-dependent SWI/SNF complex, on these demethylases for survival.”

Figure 5

Figure 5. Overexpression of KDM6s reverts sensitivity to KDM6s inhibition. *a* Western blot depicting levels of ectopic KDM6s and KDM6B in the indicated SMARCA4def cells overexpressing the indicated proteins. EV, Empty Vector. *b* Example of viability of H841 cell line, measured using MTT assays, 5 days after treatment with increasing concentrations of GSK-J4. Lines, number of viable cells relative to the total number at 0 h. Data are presented as mean ± SD from three replicate cell cultures in two experiments. *c* Distribution and mean of the EC50 from two independent experiments. Lines show mean ± SD. P-values were calculated using paired two-tailed Student's t test. *d*, Western blot depicting levels of KDM6s and SMARCA4 in the H1299 cells overexpressing the indicated proteins. *e* Viability of indicated cell lines, measured using MTT assays, 5 days after treatment with increasing concentrations of GSK-J4. Lines, number of viable cells relative to the total number at 0 h. Data are presented as mean ± SD from three replicate cell cultures in two experiments. *f* Distribution and mean of the EC50 from two independent experiments. Lines show mean ± SD. P-values were calculated using paired two-tailed Student's t test. **, P < 0.01; ***, P < 0.001; ****, P < 0.0001; ns, not significant.

SECOND REVIEW PROCESS

Reviewer #3, Replacement Reviewer for Reviewer #2 (Remarks to the Author). We thank the reviewer for having agreed to supervise the responses to reviewer No. 2 and for having reviewed our manuscript.

Comments on authors' response to prior review.

1. Addressed satisfactorily. Please replace Fig. 1d with the new data. We apologize because we believe that there has been a problem of mixed files. In the reviewed figure 1 that we sent to the journal, panel d had been replaced. Could it be that the reviewer has been looking at the figures from the first submission? Now, in the current submission we have made sure that the figure is correct.

2. The explanation provided here did not address the question “why the H1299-mutSMARCA4 cells underwent an increase in H3K27me3 concomitantly with a decrease in global H3K27ac” as the explanation addressed changes in distribution but not changes in global H3K27ac. We agree with the reviewer that the explanation may be incomplete. First, we would like to remark that, according to figure 2e, the induction of SMARCA4mut expression led to an increased in H3K27me3 but did not decrease global H3K27ac. It only affected its distribution, as explained in the previous letter. In these cells, there was a decreased in global H3K27ac, but only upon administration of SAHA.

In previous figure 3 (for review only), we showed that the ectopic overexpression of the mutant form of SMARCA4 (mutSMARCA4 dox+), in the absence of SAHA, decreased the global deposition of H3K27ac in gene bodies, without affecting the peaks at promoters. The H3K27ac peaks at promoters are even higher in the H1299-mutSMARCA4 cells upon dox administration (also shown in Suppl figure 5b) as compared to the H1299-wtSMARCA4 cells and this may be the reason why the global H3K27ac does not change (or even suffer a slight increase) (Fig. 2e). However, the global decrease in H3K27ac observed in the H1299-mutSMARCA4 cells upon administration of SAHA affects both the promoters and gene bodies (see Fig 2g and Supplementary fig.5b).

In our previous response, we stated that we were unsure about the reasons for the decrease in H3K27ac following SAHA administration in the H1299 cells overexpressing mutant SMARCA4, as compared to the H1299 parental and to other SMARCA4def parental cells. However, we hypothesized that a dominant negative function of the overexpressed mutSMARCA4 may be playing a role. Dominant negative functions of an overexpressed mutant SMARCA4 protein have been previously reported. Fernando et al. (Nat Com 2020) demonstrated that ectopic expression of some SMARCA4 aminoacid substitution mutants decrease the accessibility of SMARCA4 to the chromatin, possibly because SMARCA4 mutants can interfere with the activity of its paralog SMARCA2. This was also proposed by Hodges et al. (Nat. Struct. Mol. Biol.25, 61–72, 2018) in the context of modeling SMARCA4 heterozygous mutant expression in embryonic stem cells, which do not express SMARCA2. Somehow, the inhibition of HDACs in a dominant negative SMARCA4 activity context (complete nonfunctional SWI/SNF complex) and a deficiency in KDM6s functions may affect the equilibrium HMT/HAT/KMT/KDMs, favoring the function of the enzymes that enhance H3K27me3. Supporting this, the levels of H3K27me3 are higher in the cells treated with SAHA. We are sorry because our current data does not allow to go deeper in the molecular basis of these unexpected results. While intriguing, we believe that these are not central to the main conclusion of the current work.

Fig. 2e, the explanation provided with regard to SMARCA4 association is mediated by H3K27ac was not supported by the data in Fig. S3a (SAHA did not affect H3K27ac differentially). The treatment with SAHA in the

H1299mut cells led to a strong reduction in the global H3K27ac (heatmaps on suppl figure 4a). However, it is true that the binding of SMARCA4 in these cells does not decrease to the same extent. This may be because this is an overexpressed protein and may generate some noise. Despite this, in the normalized ChIP-seq intensities of the heatmaps (graphs above the heatmaps in Supp Fig. 4a), also in figure 2e, the numbers are quite robust and evidence a reduction in the binding of SMARCA4 in the SMARCA4 mutant cells, in parallel with a reduction in H3K27ac. While we have proposed the reduction of H3K27ac deposition as a possible explanation that prevent SMARCA4mut protein to be recruited to the DNA, we agree with the reviewer that other alternatives are possible. Because of that, we have toned down the sentence on page 8, as follows:

Previous one: *Consistent with this, the severe reduction of global H3K27ac deposition observed in SMARCA4 mutant-expressing cells following SAHA treatment, also shown by western blot (Fig. 1e), was associated with a strong reduction in the global intensity of the SMARCA4 peaks (Fig. 2e).*

New one: *Given this, the severe reduction of global H3K27ac deposition observed in SMARCA4 mutant-overexpressing cells following SAHA treatment, also shown by western blot (Fig. 1e), could be the reason for the strong reduction in the global intensity of the SMARCA4 peaks (Fig. 2e).*

Fig. 2g, it's unclear what the authors try to convey here. Based on the data, SAHA should increase KDM6A/6B in muSMARCA4 expressing cells, which would be contrary to the authors' model. The authors may discuss this.

This figure (fig 2g) depicts the IGV plots for the H3K27ac, SMARCA4, EZH2 and H3K27me3 in the H1299-SMARCA4 mut and H1299-SMARCA4wt type expressing cells, with and without treatment with SAHA. What we found is that SMARCA4 is bound to the genomic regions of KDM6A and KDM6B, in association with H3K27ac. However, there was also a concomitant increase in EZH2 occupancy in the SMARCA4mutant-expressing cells. The increase in EZH2 may underlie the reduction in the levels of KDM6s observed following the induction of overexpression of mutSMARCA4 in the H1299 cells (Suppl. Fig 3d) and the increase in global H3K27me3 upon treatment with SAHA. Taking all this into account, we believe that the observations, in the mentioned figure, are in agreement with our model.

3. Addressed satisfactorily, but there is not Supplementary Fig. 3d and 3e in the supplementary figures that match the authors' description. The reviewer is right, and we acknowledge that there is a mistake. In our previous response to the reviewer, we were referring to the data on supplementary fig.1d and 1e (now supplementary fig 2c and 2d).

4. Addressed satisfactorily.

5. Addressed, but the figure was not replaced in Fig. 2b. We are sorry because we think this is the same problem that we mentioned above (response to comment 1). In our version of the figure the panel 2b was properly replaced. Now, in the current submission we have made sure that the figure is the correct one.

However, the data as presented do not support the conclusion that "the low levels of KDM6s, basal global H3K27ac was present at a lower level in the SMARCA4def cells than in the MYCamp cells" For example, DMS273 is high in KDM6A, but low in H3K27ac. In contrast, H841 is high in KDM6B (perhaps the highest). The authors' statement on KDM6A/6B and H3K27ac is also not relevant as KDM6A/6B do not directly regulate H3K27ac, and there is no correlation between KDM6A/B and H3K27me3 (that KDM6A/6B do regulate). We also agree with the reviewer on these appreciations. The relatively low levels of H3K27ac in the DMS237, as compared to the other MYCamp cells, could be due to the variability among LC cell lines (e.g. genetic

alterations at other epigenetic factors). Despite this, the levels of KDM6A and KDM6B in these cells (DMS273) are high and comparable to that of the other MYCamp cells. Regarding the high levels of KDM6B in the H841, we want to remark that the band corresponding to the KDM6B protein is the lower one not the upper one, as the WB may suggest. This is because the antibody that had been used here is quite dirty. Now, to provide with a clearer image, we have repeated the WB using another antibody (Cell signaling JMJD3 Antibody #3457). As it can be seen in the new figure 2b (also shown in "for review fig 1", bellow) this antibody does not render the unspecific upper band. It can be observed that the levels of KDM6B in the H841 cells are low, as compared to those of the MYCamp cells. We thank the reviewer for remarking this defect that has prompted us to change the antibody and to present better-quality data.

New Figure 2b that replaces the previous one.

For review only fig 1. Western blot depicting the levels of KDM6B in the indicated cells using 2 different antibodies. The image for the new antibody, replaces the previous one in figure 2b.

Finally, as suggested by the reviewer we have removed the sentence: *“Consistent with the low levels of KDM6s, basal global H3K27ac was present at a lower level in the SMARCA4def cells than in the MYCamp cells.”* on page 6 of the manuscript.

6. Not addressed satisfactorily. Supplementary Fig. 2c (not supplementary Fig. 2b), there was a clear decrease in both KDM6A/6B by western blot. In contrast, the increase by wtSMARCA4 for KDM6A is really marginal at the best. The authors will need to address this.

We acknowledge that the increase in the KDM6s in this WB (now in supplementary Fig. 3d) is subtle. It must be considered that we are restoring the activity of SMARCA4 in a SMARCA4def cancer cell which is different as to be a cancer cell with SMARCA4 wild type. We have repeated the western-blot and, even with the new KDM6B antibody, the results are similar. We have decided to remark this fact in the manuscript by modifying the previous sentence on page 7:

Previous one: Further, the ectopic expression of the SMARCA4 wild type (H1299-wtSMARCA4 cells), but not of the mutant (H1299-mutSMARCA4 cells), triggered an upregulation of KDM6A and KDM6B and barely affected the levels of EZH2 (Fig. 2d; Supplementary Fig. 2d).

New one: Further, the ectopic expression of the SMARCA4 wild type (H1299-wtSMARCA4 cells), but not of the mutant (H1299-mutSMARCA4 cells), triggered an upregulation of KDM6A and KDM6B, albeit subtle. An opposite effect was observed for the mutant (H1299-mutSMARCA4 cells). The levels of EZH2 were unaffected (Fig. 2d; Supplementary Fig. 3d).

7. Not addressed satisfactorily. The western blot (now in supplementary Fig. 2d) clearly showed that in DMS273 cells SMARCA4 knockdown by #4 shRNA did not decrease KDM6B and both shSMARCA4s did not decrease KDM6A. Again, the authors will have to address why there is a discrepancy between their western blot and qRT-PCR and why the authors choose to believe the results of qRT-PCR over western blot.

Here, we have reselected the cells carrying the shSMARCA4 #1 and #2 using antibiotic (puromycin) and repeated the WB of KDM6A and KDM6B (with the new KDM6B antibody). We believe that the new figure shows now clearer the reduction in both KDM6s.

New Suppl. Figure 3e that replaces the previous one.

8. Addressed satisfactorily.

9. Addressed satisfactorily.

10. Addressed satisfactorily.

11. The explanation is acceptable.

12. Addressed satisfactorily.

13. Acceptable, but would be better to correlate the IC50 changes with the pathway characterized here and in particular for those MYC normal and SMARCA4 def vs. proficient cells (others as labeled) (SMARCA4, KDM6A/B, H3K27ac and H3K27me3, EZH2 etc in this panel of cell lines).

Following the reviewer's recommendations, we now provide with a western-blot of the different proteins and histone marks among the MYC normal (others) and SMARCA4 def group of LC cells (Suppl Fig 3b). The H128 cell line carries an inactivating mutation at KDM6A, that is the reason why the WB show no protein.

Fig. Suppl 3b, Western blot depicting endogenous levels of the indicated proteins in lung cancer cell lines. ACTIN and TUBULIN, protein-loading controls. The H228 cells do not have KDM6A protein due to a genetic inactivating alteration at the KDM6A gene

To include this information, we have modified the previous sentence on page 8, to this new one:

“Searching the Cancer Cell Line Encyclopedia (CCLE) we found significantly lower levels of expression of several histone demethylases (KDMs), including KDM6A and KDM6B, in SMARCA4def compared with MYCamp cells or with LC cells that are wild type for SMARCA4 and MYC (Fig.2a Supplementary Fig. 3a). We validated our observations in a panel of cell lines at the mRNA and protein levels (Fig.2c-d; Supplementary Fig. 3b).”

14. This reviewer agrees with the prior reviewer that the difference in marginal and in particular in DMS273 cells and shKDM6B in H460 cells. The conclusion is an over interpretation of the data presented.

We acknowledge that the differences are marginal for the some of the cells and some of the shKDM6B. We did state that in the manuscript (page 11), following sentence:

“Moreover, the lower levels of KDM6A, and to a lesser extent of KDM6B, in the MYCamp cells, sensitised the cells to the treatment with GSK-J4 (Fig. 4e-f).”

We have modified the sentence as follows:

“Moreover, the lower levels of KDM6A, and to a lesser extent of KDM6B, in the MYCamp cells (especially in the H82 and H460 cells), sensitised the cells to the treatment with GSK-J4 (Fig. 4e-f).”

In the DMS273 cells the observations are also marginal for the sh63 but not for the sh64 expressing cells (reduced the EC50 by half). However, in these cells the effectiveness of sh63 in reducing KDM6B proteins levels was lower as compared to the sh64 or to the other cell lines (see figure S5b). It also needs to be considered that the levels of KDM6A in these cells are among the highest (fig 1b).

15. Addressed satisfactorily.

FIRST REVIEW PROCESS (Reviewer #3)

Changes to comments raised by Reviewer #3:

Major comments:

1. The rationale for targeting KDM6A/6B in KDM6A/6B low SMARCA4 inactivated cells is missing. What is the mechanism that the authors propose to account for these findings? At the minimum, inducible knockdown of KDM6A/6B in isogenic SMARCA4 proficient vs. deficient cells should be used as a control for on target effects.

Conversely, if the model as proposed by the authors are correct, ectopic expression of KDM6A/6B in SMARCA4 deficient cells should rescue the growth inhibition induced by GSK-J4.

3. There is no control for off-target effects of GSK-J4 in their experimental design and this is important given GSK-J4 suppresses a number of KDMs. For example, it would be informative to test GSK-J4 in KDM6A/B knockout cells.

Response to comments 1 and 3:

We agree with the reviewer that the ectopic expression of KDM6A/B will provide with important clues about the mechanism underlying the specific sensitivity of SMARCA4def cells to KDM6i. The same comment has been prompted by reviewer No.1. Now, we have overexpressed KDM6A and KDM6B in some SMARCA4def LC cancer cells and treated them with GSK-J4. Please see the data in our response to reviewer No. 1. These new pieces of data further support the on-target effect of GSK-J4 for KDM6 in these cells.

On the other hand, knockdown (not inducible) of KDM6A and KDM6B was performed in a panel of SMARCA4def and SMARCA4proficient LC cells. SMARCA4def cells carrying shRNAs against KDM6A or KDM6B were not viable (Fig 4d), attesting that inhibition of KDM6s is lethal in the SMARCA4def cells. In contrast, in the cells that carry MYC-amplification the down-regulation of each KDM6 barely affected cell growth. We would expect the same to happen if, instead of shRNAs, we had used CRISPR/CAS9 to KO the KDM6s.

2. The findings related to HDAC inhibition and EZH2 inhibition are contrary to a number of publications in the literature (e.g., Wang et al, J Path, 2017; Chan-Penenbre et al, Mol Cancer Ther, 2017, Wang et al, Mol Cancer Ther, 2018). The authors completely ignored these prior findings and did not provide any explanation.

We acknowledge the comment of the reviewer about that there are previous works that provide with evidence about the role of EZH2 inhibition in suppressing the growth of small cell carcinoma of the ovary, hypercalcemic type (SCCOHT) cell lines. While in our current work the SMARCA4 deficient tumors did not show specific sensitivity to GSK126 compound, at least as compared to the MYCamp cells, it is true that few SMARCA4def cell were tested. The marked resistance of these two cells to GSK126 may not be extended to all SMARCA4def lung cancer cells. It is also possible that, as compared to lung cancer cells with SMARCA4 mutations, SCCOHTs show sensitivity to GSK126 due to their genetic characteristics. These are genetically quite stable tumors and do not tend to carry many mutations in other oncogenes or tumor suppressor genes. In a recent study, EZH2 inhibitors were tested as single agents or before chemotherapy in mice with orthotopic Kras-driven NSCLC grafts. These tumors displayed sensitivity to GSK126 but also amplify an inflammatory program that allowed tumor cells to overcome the initial GSK126 antiproliferative effects (*Serresi M, et al. Exp Med. 2018;215(12):3115-35*).

Now, we have included this information, on page 6 of the manuscript. The new sentence reads as follows:

“Previous studies have shown that GSK126 alone or combined with HDAC inhibitors suppress the growth of the small cell carcinoma of the ovary, hypercalcaemic type (SCCOHT)²⁴⁻²⁵. ”

Two additional references have also been included (24 and 25).

24.-Chan-Penebre, E., et al. Selective killing of SMARCA2- and SMARCA4-deficient small cell carcinoma of the ovary, hypercalcemic type cells by inhibition of EZH2: in vitro and in vivo preclinical models. *Mol Cancer Ther.*16, 850-860 (2017).

25.-Wang, Y., et al. Histone deacetylase inhibitors synergize with catalytic inhibitors of EZH2 to exhibit antitumor activity in small cell carcinoma of the ovary, hypercalcemic type. *Mol Cancer Ther.* 17, 2767-2779 (2018).

4. Given the differential responses to SAHA, the changes in various factors/markers could simply be due to changes in cell growth kinetics/cell cycle distribution that are known to regulate relevant activities such as EZH2. Fig. 1: d, cell cycle distribution should be examined.

The reviewer raises an important point since EZH2 levels are known to be decreased during cell growth inhibition. Now, we have performed flow cytometry in a panel of SMARCA4def and MYCamp cells from figure 1d and 1f. The results show that, in the MYCamp cells, SAHA blocks cell cycle progression at the G0/G1 or subG1 phases whereas no changes were observed in the SMARCA4def cells. An increased in apoptosis (PARP cleavage determination) was also evident in some of the cells (especially DMS273 and H446), as opposed to the SMARCA4def cells. Here, we want to remark that we had to change the concentration of SAHA, from 1 to 0.5 μ M. With 1 μ M of SAHA all the MYCamp cells quickly disappeared from the plates and it was impossible to perform the flow cytometry experiments.

This new information has been added on page 5 of the manuscript, in Supplementary Fig 1b and in the methods section. The previous supplementary figures 1c-e have been moved to a new supplementary figure (Suppl. Fig.2). The remaining supplementary figures have been renamed, accordingly.

The new sentence on page 5:

“Flow cytometric analysis showed that, in the MYCamp cells, SAHA blocked cell cycle progression at the G0/G1 or SubG1 phases and increased apoptosis, whereas no changes were observed in SMARCA4def cells (Supplementary Fig. 1b-d).”

The new supplementary figure:

Additional figure (Suppl. Figure 1b-d). Supplementary Figure 1 b, SAHA induced G1 phase arrest in the cell cycle, in MYCamp but not in SMARCA4def cells. Cell cycle distribution was obtained by PI staining and flow cytometry analysis in cells following treatment with 0.5 µM of SAHA for 48 hrs. c, Bar graphs depicting the percentage of the indicated cell in each of the cell cycle phases with and without SAHA treatment. d, Western blotting depicting the cleavage of PARP in the indicated cells, following the treatment with SAHA or vehicle.

Fig. S1b, why and how did SAHA change (decrease or increase) EZH2 expression? What we observed was that, in most of the SMARCA4def cells the treatment with SAHA did not increase EZH2 levels (Fig 1f). However, we found that H23 was an exception (previous Supp fig 1b, now supp fig 2a). We have now, clarified this in the text (page 6):

Previous sentence: “The administration of SAHA did not alter the levels of EZH2 in most SMARCA4def cells, indicating that overactivation of the methyltransferase activity of EZH2 is unlikely to account for the defects in H3K27me3 triggered by SAHA in these cells (Fig. 1f; Supplementary Fig. 2a)”

New sentence: “The administration of SAHA did not alter the levels of EZH2 in most SMARCA4def cells (except in the H23 cells which showed an increase), indicating that overactivation of the methyltransferase activity of EZH2 is unlikely to account for the defects in H3K27me3 triggered by SAHA in these cells (Fig. 1f; Supplementary Fig. 2a)”

e, there is no decrease in H3K27ac in wtSMARCA4 re-expressed cells, indicating off-target effects of SAHA. In figure 1e, a decrease in H3K27ac in wtSMARCA4 cells was not expected. All the cancer cells tested, in both the SMARCA4def and the MYCamp groups, were capable to increase H3K27ac upon administration of SAHA (see fig. 1d & fig 1f).

f, why SAHA decreases EZH2 and GSK126 further decreases EZH2? Regarding the question of the reviewer as to why SAHA decreases EZH2 and GSK126 decreases EZH2 further (Fig 1f and Supp Fig 2a), we also agree this is intriguing and still do not have a definitive answer. The decrease in the levels of EZH2 upon administration of SAHA was strong, reproducible, and specific of the MYCamp cells (not of the SMARCA4def cells). Previous studies have indicated that EZH2 interacts with class I HDACs, HDAC 1 and 2, through another PRC2 protein, EED (*van der Vlag J, Otte AP. Nat Genet 1999; 23: 474–8*) and that transcriptional repression by EZH2 requires the activity of the HDAC (*Varambally S, et al. Nature 2002;419:624–9*). Moreover, hydroxamate histone deacetylase inhibitors LBH589 or LAQ824 have shown to deplete the protein levels of EZH2 and EED in the cultured and primary human acute leukemia cells (*Fiskus W, Pranpat M, Balasis M et al. Mol Cancer Ther 2006; 5: 3096–104*).

According to these previous works, the inhibition of HDACs using SAHA would suppress transcriptional repression by EZH2 and the protein levels of EZH2 itself. The treatment with GSK126 should also enhance this inhibition. It is intriguing that, according to our current observations, the decrease in EZH2 levels by HDACi does not occur in SMARCA4def cells, indicating that the molecular basis for the decreased in EZH2 by SAHA requires an intact SWI/SNF complex.

5. The authors observed upregulation of H3K27me3 by SAHA in SMARCA4 deficient cells, and hypothesize that this might be the reason of resistance to SAHA. However, restoration of H3K27me3 by GSK-J4 failed to re-sensitize SMARCA4 inactivated cells to SAHA, which argues against this model. In addition, GSK126 obviously caused resistance to SAHA in DM273 cells, while the authors claimed GSK126 did not affect sensitivity and EZH2 is not implicated. At least, it is possible that EZH2 participates the repression of KDM6A/B caused by mutSMARCA4.

We apologize because we did not intend to conclude that the high levels of H3K27me3 and the global increased of H3K27me3 is the reason for the resistance to SAHA in SMARCA4def cancer cells. Our model supports that, in the absence of a functional SMARCA4, there are defects in the levels and in the activity of KDM6s which affects the dynamics of the H3K27me3 mark, following administration of SAHA. In fact, our experimental data using shKDM6A and shKDM6B supports that a deficient KDM6B is responsible for the refractoriness to SAHA. We have checked throughout the manuscript to modify the sentences that may lead to the conclusion that the upregulation of H3K27me3 by SAHA is the reason for the refractoriness to this compound.

In the abstract

Previous

“This is associated with impaired transactivation and significantly reduced levels of the histone demethylases KDM6A/UTX and KDM6B/JMJD3, and with a strong dependency on these histone demethylases, so that its inhibition compromises cell viability in the SMARCA4-mutant cells “

Current

“ SMARCA4-mutant cells also show an impaired transactivation and significantly reduced levels of the histone demethylases KDM6A/UTX and KDM6B/JMJD3, and a strong dependency on these histone demethylases, so that its inhibition compromises cell viability.

The following paragraph on page 9:

“Next, using shRNAs, we downregulated KDM6A and KDM6B expression in different MYCamp cells (Fig, 3a; Supplementary Fig. 5b), and noted that, mimicking the behavior of the SMARCA4def cells, the reduction in KDM6B levels suppressed the ability to reduce the global level of H3K27me3 deposition (Fig. 3b) and to inhibit

cell growth (Supplementary Fig. 5c) by SAHA. Conversely, the depletion of KDM6A did not affect these characteristics. The administration of the small molecule compound GSK-J4, a very specific inhibitor of KDM6A/KDM6B, reverted the sensitivity to SAHA in a dose-dependent manner (Fig. 3c-d) and prevented the SAHA-triggered global decrease in H3K27me3 in both KDM6s-depleted cells (Fig. 3b).

These findings suggest that the low levels of KDM6B account for the resistance of the SMARCA4def cells to growth inhibition by SAHA, and hint at a more widespread role for KDM6B in the global removal of H3K27me3.”

Has been modified to this new one:

Next, using shRNAs, we downregulated KDM6A and KDM6B expression in different MYCamp cells (Fig, 3a; Supplementary Fig. 6b), and noted that, mimicking the behavior of the SMARCA4def cells, the reduction in KDM6B levels, but not of KDM6A, suppressed the ability to inhibit cell growth (Supplementary Fig. 6c) by SAHA. The depletion of KDM6B, but not of KDM6A, also prevented the decreased of the global level of H3K27me3 deposition by SAHA (Fig. 3b), hinting at a more widespread role for KDM6B in the global removal of H3K27me3. The administration of the small molecule compound GSK-J4²⁸, a very specific inhibitor of KDM6A/KDM6B, has similar effects, reverting the sensitivity to SAHA in a dose-dependent manner (Fig. 3c-d) and prevented the SAHA-triggered global decrease in H3K27me3 in both KDM6s-depleted cells (Fig. 3b).

These findings suggest that a deficiency in KDM6B account for the resistance of the SMARCA4def cells to growth inhibition by SAHA.

The following sentence in the discussion section:

“Here, we found that a deficient KDM6B is responsible for the refractoriness to SAHA and for the global increase in H3K27me3 upon administration of SAHA in SMARCA4def cells. This is consistent with a broader role for KMD6B in H3K27me3 deposition than that of KDM6A³²”

has been modified to this new one:

“Here, we found that a deficient KDM6B is responsible for the refractoriness to SAHA in SMARCA4def cells. Further, a deficient KDM6B also accounts for the global increase in H3K27me3 upon administration of SAHA, which is consistent with a broader role for KMD6B in H3K27me3 deposition than that of KDM6A³⁴”

Here the reviewer also comments on the refractoriness to SAHA that the EZH2 inhibitor compound, GSK126, triggered in the MYCamp cells DMS273 (Supplementary Figure 2c). We agree that this result is confusing. The administration of SAHA and GSK126 strongly decreased the levels of EZH2 in all the cells (Fig 1f) and, in our hands, the addition of GSK126 did not sensitize the SMARCA4def cells to SAHA. Thus, given that this was the message that we wanted to highlight here and that we have not explanation for the observation in the MYCamp (DMS273) cells we have decided to remove the two MYCamp cells from the Supplementary Fig 1d and e. We believe that this data generates more questions than answers and deserves of more experimental investigation. Accordingly, we have also modified the sentence on page 6: “The administration of GSK126, alone or in combination with SAHA, did not reduce the proliferation or viability of any group of cells (Supplementary Fig. 2b-d).” to this new one: “The administration of GSK126, alone or in combination with SAHA, did not reduce the proliferation or viability of the SMARCA4def cells (Supplementary Fig. 2b-d).”

Finally, the reviewer comments on the possibility that EZH2 participates in the repression of KDM6A/B in the SMARCA4def cells. In the ChIP-seq of EZH2 (Fig 2e) it can be observed an increase in EZH2 occupancy in the KDM6s promoters following overexpression of the mutant SMARCA4 in the H1299 cells but not in the parental

(dox-) cells (that also lack SMARCA4). So, in this cell model we believe that EZH2 activity is repressing the already low levels of KDM6s.

We have performed a WB of KDM6A, KDM6B and EZH2 in MYCamp and SMARCA4 cells treated with the EZH2 inhibitor GSK126. No changes in the levels of none of these proteins was observed. This suggest that EZH2 methyltransferase activity ma not responsible for the reduced levels of KDM6A/6B in the SMARCA4def cells.

For review only fig 2. Western blot depicting the levels of KDM6A and KDM6B in the indicated cells and with or without GSK126 treatment. In the H446 no KDM6s proteins can be detected because the total protein loading was too low.

6. Based on Fig. 2e and g, increased binding of EZH2 and H3K27me3 by mutSMARCA4 in the KDM6B promoter could be the reason why KDM6A/B expression was decreased in mutSMARCA4 cells. Did the authors check the role of EZH2 in mutSMARCA4 mediated KDM6A/B decrease? As mentioned above, it is highly possible that EZH2 binding to the KDM6s promoters is the reason why KDM6s decreased in the H1299 cells upon induction of mutSMARCA4.

In fact, this is mentioned in the manuscript, on page 9, in the sentence:

“The increase in EZH2 in the promoter of these genes is consistent with a lack of transcriptional activation of these KDMs in the H1299-mutSMARCA4 cells (Fig. 2d)”

This sentence has now been slightly modified to highlight that there is also some repression.

“The increase in EZH2 in the promoter of these genes is consistent with a lack of transcriptional activation and even some transcriptional repression of these KDMs in the H1299-mutSMARCA4 cells (Fig. 2d; Supplementary Fig. 3d)”

In addition, why are KDM6A/B upregulated by wtSMARCA4? We believe that in the absence of SMARCA4 there is not chromatin remodelling to make chromatin accessible and, therefore, only basal (low) levels of these proteins are expressed. The reconstitution of SMARCA4 expression (wtSMARCA4) would allow for chromatin remodelling to take place in these genes and, therefore, for transcriptional regulation/activation of the KDM6s.

Why was SAHA not effective in decreasing H3K27ac in wtSMARCA4 in Fig. 1e and Fig.S3a. We did not expect SAHA to decrease H3K27ac in wtSMARCA4. We expected H3K27ac to increase while H3K27me3 decrease.

Minor:

1. Fig. 5d, DMS273X tumors should be included as controls. As recommended by the reviewer, we have now included the immunostaining of the DMS273X tumor in previous figure 5d (now figure 6d).

Figure 6d. Representative immunostaining of H3K27me3 in tumours from the indicated cells and treatments. Scale bars, 25 μ m. Below, distribution of H3K27me3 staining among tumours (three tumours per cell line and condition) from the DMS273X, H841X and DMS114X tumours treated with vehicle or GSK-J4. Low (intensity values 1 & 2); high (intensity values 3 & 4) (Supplementary Table 4). Fisher's Exact test.

2. Fig. 6: b, the quality of the image is poor. We now provide with larger and improved images of the clonogenic assays, on figure 6b.

6d, Why GSK-J4 did not increase H3K27me3? The effect of GSK-J4 on global H3K27me3 increase was clear in OVCAR-8 (SMARCA4wt) and OVA259L (SMARCA4def) cells. However, this was not observed in OVA250L cells. We want to point out here that the increase in H3K27me3 after administration of GSK-J4 is evident in the IHC of this tumor when it is implanted orthotopically in mice (Fig. 7h). The failure to observe an increase H3K27me3 in the western-blot of the OVA250L cells we believe is because these cells are very sensitive to GSK-J4 (perhaps the most sensitive among all the SMARCA4def cells tested in our work). This fact led to strong cell death in cells at 48 h (used in most of the assays). Now, we have repeated the Western-Blot (see figure below) using 24hr treatment with GSK-J4, for the OVA250L cells. The increase in H3K27me3 after GSK-J4 administration is now evident. This has now been indicated in the figure legend.

Figure 7d. Western blot of the endogenous levels of the indicated proteins and cancer cells. TUBULIN, protein-loading control (48hr treatments, except 24hr in the GSK-J4 for the OVA250L cells).

6f, the tumor volume does not appear to be consistent with the image. We believe this is because the first tumor sample in the GSK-J4-treated tumor lane has an attached blood clot (can be seen to the right of the image). This clot was not considered when measuring the volume of the tumor sample.

For review only fig 3. Gross pathological photographs at necropsy of two of the ovarian tumours (the larger ones) that arose in mice treated with vehicle (left) or GSK-J4 (right) (from Figure 7f). As can be seen, the tumor sample on

the right has an attached blood clot. Therefore, the volume is much smaller than that of the tumor on the left.

6h, data should be quantified. We have added a new supplementary table that provide with the quantification of the immunostainings for H3K27me3, including the individual values of each replicate and of the three different evaluators. A sentence in the legend of figure 3h has been included that refers to this new table: “The quantification of the immunostaining is provided in Supplementary Table 4.”

Supplementary Table 4. Distribution of H3K27me3 staining among tumours (three replicates per cell line and conditic from the DMS273X, H841X, DMS114X and OVA250X tumours treated with vehicle or GSK-J4. Low (intensity values 1 & high (intensity values 3 & 4).

	Repli	Eval 1		Eval 2		Eval 3		Mean		Categories	
		H3K27	H3K27	H3K27	H3K27	H3K27	H3K27	H3K27	H3K27	H3K27	H3K27
		Vehic	GSK	Vehic	GSK	Vehic	GSK	Mean-V	Mean-GS	Catego Vehic	Catego GSK
DMS114	R1	1	3	1	2	1	1	1	2	Low	Low
	R2	2	4	2	3	2	3	2	3.3	Low	High
	R3	2	4	2	4	2	4	2	4	Low	High
H841X	R1	1	3	1	3	1	3	1	3	Low	High
	R2	3	4	3	3	3	3	3	3.3	High	High
	R3	2	4	2	4	2	3	2	3.6	Low	High
DMS273	R1	2	3	1	2	1	2	1.3	2.3	Low	Low
	R2	1	3	2	3	2	3	1.6	3	Low	High
	R3	2	4	2	3	2	4	2	3.6	Low	High
OVA250	R1	2	3	1	3	2	3	1.6	3	Low	High
	R2	2	4	1	4	2	4	1.6	4	Low	High
	R3	2	3	2	4	3	4	2.3	3.6	Low	High

3. Fig. S5: the data as presented would only support KDM6A, but not KDM6B. In Fig 5S (now Fig. S6) our conclusions from the data are that KDM6B depletion mimics the response of SMARCA4-deficient cells to SAHA. In our view there are the following experimental observations that supports it:

-First, we found that the mRNA levels of the KDM6B were inversely correlated with the EC50 to SAHA (Supplementary Fig. 6a).

-Second: using shRNAs, we downregulated KDM6A and KDM6B expression in different MYCamp cells (Fig, 3a; Supplementary Fig. 6b), and noted that, mimicking the behavior of the SMARCA4def cells, the reduction in KDM6B levels suppressed the ability to reduce the global level of H3K27me3 deposition (Fig. 3b).

-Third: the same shRNAs against KDM6A and KDM6B in different MYCamp cells suppressed the ability of SAHA to inhibit cell growth, only in the KDM6B (Supplementary Fig. 6c).

4. Fig. S6: the differences in panel d (in the figure) are marginal at the best. Yes, the reviewer is right. The differences in the EC50 to GSK-J4 following the restitution of wtSMARCA4 in the MTT and clonogenic assays is minor (now Supp. Fig 7c-d). This is also in line with the subtle increase in KDM6A and KDM6B observed in figure 4d. It must be considered that we are restoring the activity of SMARCA4 in a SMARCA4def cancer cell which is different as to be a cancer cell with SMARCA4 wild type.

5. The authors should clearly label the cell lines as MYCamp or SMARCA4 mutant in every figure.

We thank the reviewer for this suggestion. We have gone through all the figures and have made sure that it is indicated, at least once in every figure, whether the cells are MYCamp or SMARCA4def. The following figures have been updated: Fig3c, Fig 6b, Suppl Fig.8, Suppl Fig 9.

6. There are many data as described in the text were not presented and in particular those related to database mining (in addition to mis-labeling). They include S3c, S2d-e, S6b, S6e etc. We are sorry because we think there has been some problem with the files. We believe that the reviewer must have been using the old version figures. In our corrected version all the figures contain the database mining. The mis-labeling referred by the reviewer may also obey to this misunderstanding. Nevertheless, we have checked all the figures and try

to provide with more details in some of the supplementary figures. However, we did find one figure that was named in the text and was not included in the figures, that was Supplementary Fig 7e. we have now included this figure that was previously included only as a figure for reviewed purpose, only.

Figure 7e, Dependency of different cancer cell lines for each of the indicated KDM. Blue, SMARCA4def cell lines; Red, MYCamp cell lines and grey, cancer cell lines that are wt for both SMARCA4 and MYC. (Achilles project: <https://depmap.org/portal/achilles/>)

REVIEWERS' COMMENTS

Reviewer #1 (Remarks to the Author):

The authors have addressed my comments.

Reviewer #3 (Remarks to the Author):

The authors have substantially addressed the concerns raised by the reviews and the manuscript is improved on both data quality and interpretation. This is much appreciated by this reviewer. To increase clarity and accuracy, the authors may consider revise some of the statements and descriptions by text changes:

1. SAHA decreased H3K27ac (e.g., in Fig. 1e), which correlates with an increase in H3K27me3 in mutSMARCA4 cells. This needs to be discussed.
2. Loss of EZH2 expression in MYCamp cells could be due to either cell cycle arrest and/or apoptosis. This can be discussed.
3. Statements related to changes in expression between MYCamp and SMARCA4def can be modified to be accurate. For example, KDM6A/6B are both expressed at higher levels in A427 SMARCA4def cells and KDM6A is expressed at a higher level in DMS114 cells (compared with MYCamp cells).
4. Minor errors should be corrected throughout the manuscript that includes figure references and missing statistical analysis (e.g., Fig. 2a KDM6A and KDM1A).

REVIEWERS' COMMENTS:

Reviewer #1 (Remarks to the Author):

The authors have addressed my comments.

We thank the reviewer for having reviewed our work.

Reviewer #3 (Remarks to the Author):

The authors have substantially addressed the concerns raised by the reviews and the manuscript is improved on both data quality and interpretation. This is much appreciated by this reviewer. To increase clarity and accuracy, the authors may consider revise some of the statements and descriptions by text changes:

1. SAHA decreased H3K27ac (e.g., in Fig. 1e), which correlates with an increase in H3K27me3 in mutSMARCA4 cells. This needs to be discussed.

It is true that we had answered to this same question prompted by the reviewer (previous question No. 2) but we did not discuss the findings in the manuscript. In our previous response we stated: *“we were unsure about the reasons for the decrease in H3K27ac following SAHA administration in the H1299 cells overexpressing mutant SMARCA4, as compared to the H1299 parental and to other SMARCA4def parental cells. However, we hypothesized that a dominant negative function of the overexpressed mutSMARCA4 may be playing a role. Dominant negative functions of an overexpressed mutant SMARCA4 protein have been previously reported. Fernando et al. (Nat Com 2020) demonstrated that ectopic expression of some SMARCA4 aminoacid substitution mutants decrease the accessibility of SMARCA4 to the chromatin, possibly because SMARCA4 mutants can interfere with the activity of its paralog SMARCA2. This was also proposed by Hodges et al. (Nat. Struct. Mol. Biol.25, 61–72, 2018) in the context of modeling SMARCA4 heterozygous mutant expression in embryonic stem cells, which do not express SMARCA2. Somehow, the inhibition of HDACs in a dominant negative SMARCA4 activity context (complete nonfunctional SWI/SNF complex) and a deficiency in KDM6s functions may affect the equilibrium HMT/HAT/KMT/KDMs, favoring the function of the enzymes that enhance H3K27me3. Supporting this, the levels of H3K27me3 are higher in the cells treated with SAHA. We are sorry because our current data does not allow to go deeper in the molecular basis of these unexpected results. (...)”*.

Now we have added the following sentence in the manuscript, on page 6:

“A dominant negative function of an overexpressed mutant SMARCA4 protein may underlie this effect in the H1299-mutSMARCA4 cells.”

2. Loss of EZH2 expression in MYCamp cells could be due to either cell cycle arrest and/or apoptosis. This can be discussed.

The reviewer is right since a down-regulation of EZH2 is associated to cell growth inhibition and apoptosis mechanisms. Now, we have modified the previous sentence on page 6:

“In the MYCamp cells, SAHA treatment alone reduced the EZH2 levels, the effect being enhanced by the addition of GSK126, an inhibitor of the enzymatic activity of EZH2.”

To this new one:

“In contrast, treatment with SAHA alone reduced EZH2 levels in the MYCamp cells, consistent with the fact that SAHA triggers inhibition of cell cycle progression and increases apoptosis in cancer cells with this genetic context.”

3. Statements related to changes in expression between MYCamp and SMARCA4def can be modified to be accurate. For example, KDM6A/6B are both expressed at higher levels in A427 SMARCA4def cells and KDM6A is expressed at a higher level in DMS114 cells (compared with MYCamp cells).

The reviewer is right, and we acknowledge that the sentence needs to be more accurate. We have now changed the previous sentence on page 7:

“We validated our observations in a panel of cell lines at the mRNA and protein levels (Fig.2c-d; Supplementary Fig. 3b).”

To this new one:

“We validated our observations in a panel of LC cell lines at the mRNA and protein levels in which, with few exceptions, SMARCA4def LC cells carry lower levels of KDM6A and KDM6B as compared to the MYCamp cells (Fig.2b-c; Supplementary Fig. 3b).”

4. Minor errors should be corrected throughout the manuscript that includes figure references and missing statistical analysis (e.g., Fig. 2a KDM6A and KDM1A).

The reviewer is right. We have corrected the above-mentioned error and have looked throughout the entire manuscript and correct some mistakes, especially in figure references.